



# Snowpack and firn densification in the Energy Exascale Earth System Model (E3SM) (version 1.2)

Adam M. Schneider[1], Charles S. Zender[1], and Stephen F. Price[2]

[1]University of California, Irvine, Department of Earth System Science, Croul Hall, Irvine, CA 92697-3100
[2]Fluid Dynamics and Solid Mechanics Group, Los Alamos National Laboratory, Los Alamos, NM 87545, U.S.A.

**Correspondence:** Adam Schneider (amschnei@uci.edu)

**Abstract.** Earth's largest island, Greenland, and the Antarctic continent are both covered by massive ice sheets. A large fraction of their surfaces consist of multi-year snow, known as firn, which has undergone a process of densification since falling from the atmosphere. Until now this firn densification has not been fully accounted for in the U.S. Department of Energy's Energy Exascale Earth System Model (E3SM). Here, we expand the E3SM Land Model (ELM) snowpack from 1 m to up to 60 m to

enable more accurate simulation of snowpack evolution. We test four densification models in a series of century-scale land surface simulations forced by atmospheric re-analyses, and evaluate these parameterizations against empirical density-versus-depth data. To tailor candidate densification models for use across the ice sheets' dry-snow zones, we optimize parameters using a regularized least squares algorithm applied to two distinct stages of densification. We find that a dynamic implementation of a semi-empirical compaction model, originally calibrated to measurements from the Antarctic peninsula, gives results more

consistent with ice core measurements from the cold, dry snow zones of Greenland and Antarctica, compared to when using the original ELM snow compaction physics. In its latest release, the Community Land Model (CLM) (version 5) provides updated snow compaction physics that we test in ELM, resulting in top 10 m firn densities that are in better agreement with observations than densities simulated with the semi-empirical model. Below 10 m, however, the semi-empirical model gives results that more closely match observations, while the current CLM(v5) compaction physics predict firn densities that increase too slowly with

depth and are thus unable to simulate pore close off (a phenomenon of particular interest to paleoclimate studies). Because snow and firn density play roles in snowpack albedo, liquid water storage, and ice sheet surface mass balance, these improvements will contribute to broader E3SM efforts to simulate the response of land ice to atmospheric forcing and the resulting impacts on global sea level.

## 1 Introduction

Mass loss from the surface of the Greenland Ice Sheet (GrIS) is a major contributor to mean sea level rise (SLR). Recent estimates indicate that from 1991 to 2015, an average of 0.47 (+/- 0.23) mm yr$^{-1}$ of barystatic SLR is attributable to changes in GrIS surface mass balance (SMB), defined as the annual rate of surface accumulation, due to snowfall, minus ablation (van den Broeke et al., 2016). While historical reconstructions of GrIS SMB apply regional climate models forced at their boundaries by atmospheric reanalyses (Noël et al., 2018; Fettweis et al., 2017; van Angelen et al., 2014), predictions of



future GrIS SMB contribution to SLR requires global, coupled Earth System Models (ESMs) coupled to dynamic ice sheet components (Lenaerts et al., 2019; Muntjewerf et al., 2020). As ESMs are progressively tailored to study ice sheet-climate interactions and feedbacks, they require a complex snowpack model that can simulate the essential processes directly involved in the surface mass and energy budgets (Fyke et al., 2018; Vizcaino, 2014).

In the Energy Exascale Earth System Model (version 1) (Golaz et al., 2019), snowpack conditions on glaciers and ice sheets

(i.e., land ice) are simulated within its land component, the E3SM Land Model (ELM). Although there are complex snowpack models capable of simulating alpine snow conditions, a complete snow compaction model suitable for use within a global land surface model that includes snow on ice sheets, like the ELM, does not yet exist. Recently, van Kampenhout et al. (2017) expanded the shallow (1 m maximum snow water equivalent, or SWE hereafter) snowpack model used in version 4 of the Community Land Model (CLM) and implemented into version 5 the one-dimensional overburden pressure compaction model

from Vionnet et al. (2012) (Lawrence et al., 2019). Despite these advances, top 1 m ice sheet densities, as well as characteristic firn depths, simulated by the CLM are barely correlated ($R^2 = 0.15$) with observations. These snowpack models, which are commonly used in ESMs to simulate seasonal snow cover, do not include the essential processes governing densification rates for higher densities that are typical of multi-year snow (i.e., firn). Although firn densification models can produce plausible density versus depth relationships for the region for which they are calibrated, they do not result in consistent depth integrated

porosities due to their incomplete representations of micro-physical processes (Lundin et al., 2017). As it pertains to the ELM, the first steps toward improving firn processes relevant to SMB are enabling and evaluating firn densification in dry-snow zones, where we can neglect the effects of liquid water (Steger et al., 2017). This is a major simplification since densification in percolation zones, and the resulting formation of ice lenses and aquifers, is an ongoing area of research (Miège et al., 2016; Munneke et al., 2014; Koenig et al., 2014; Forster et al., 2014).

The purpose of this study is to enable, evaluate, and improve the simulation of dry firn densification in the ELM. To this end, we expand the current shallow snowpack model in ELM and simulate snow accumulation, aging, and compaction on land ice. We compare our simulation results against empirical data from firn core density measurements and those predicted by Herron and Langway (1980). We analyze the resultant mechanics of simulated firn by approximating vertical strain rates from steady-state density profiles and accumulation rates. We then optimize the firn densification expression(s) against observations

to obtain the parameterization that will be used in future versions of ELM and E3SM.

## 2  Background

Relevant modeling approaches can be divided into two general applications: seasonal snowpack models, in which (relatively low density) snow fractional compaction rates are calculated using the finite-element method (Podolskiy et al., 2013); and firn models, which provide either an explicit density versus depth relationship or an explicit densification rate given as a function of

overburden pressure, local temperature, and bulk density. In this study, we seek to validate the representation of both snowpack and firn densification in the ELM.


## 2.1 Snowpack in Earth system and land surface models

Snow compaction in Earth system and land surface models is routinely simulated using one-dimensional parameterizations of the general form

$$\dot{\epsilon} = -\frac{\sigma}{\eta}, \tag{1}$$


which relates vertical strain rates $\dot{\epsilon}$ to overburden pressures $\sigma$ by a viscosity function $\eta$. As in the land component CLM(v4) of the Community Earth System Model (CESM) (Lawrence et al., 2011), the original ELM (version 1) further represents snow strain (i.e., fractional compaction) rates as the sum of three terms representing *destructive metamorphism*, *overburden pressure*, and *melt* (Anderson, 1976; Oleson et al., 2013). Densification due to destructive metamorphism (denoted by the subscript

"dm"), which includes the settling of snow grains as they age, is calculated as a temperature ($T$) dependent, piecewise-defined function of density ($\rho$), expressed by its engineering strain equivalent

$$\left(\frac{1}{\Delta z}\frac{\partial \Delta z}{\partial t}\right)_{\mathrm{dm}} = \begin{cases} -c_3 \exp[c_4(T - T_f)], & \text{if } \rho < \rho_{\mathrm{dm}} \\ -c_3 \exp[c_4(T - T_f) - c_5(\rho - \rho_{\mathrm{dm}})], & \text{if } \rho \geq \rho_{\mathrm{dm}} \end{cases}, \tag{2}$$

for each vertical layer with thickness $\Delta z$, constants $c_3 = 2.78 \times 10^{-6}$ s$^{-1}$, $c_4 = 0.04$ K$^{-1}$, $c_5 = 46 \times 10^{-3}$ m$^3$kg$^{-1}$, $T_f = 273.15$ K, and a density threshold $\rho_{\mathrm{dm}}$ (100 kg m$^{-3}$) above which the densification rate tapers off. At any given depth $z$, the

fractional compaction rate due to overburden (load) pressure (denoted by the subscript "op") is calculated from the integrated column density above $z$, denoted by $\sigma$, by expanding eq. (1) as

$$\left(\frac{1}{\Delta z}\frac{\partial \Delta z}{\partial t}\right)_{\mathrm{op}} = \frac{-\sigma}{\eta_0 \exp\left[c_1(T_f - T) + c_2\rho\right]}, \tag{3}$$

for each vertical layer with thickness $\Delta z$, constants $c_1 = 0.08$ K$^{-1}$, $c_2 = 0.023$ m$^3$kg$^{-1}$, and a viscosity coefficient $\eta_0 = 9 \times 10^5$ kg s$^{-1}$ m$^{-2}$. The snowpack model includes a one dimensional temperature model, where temperatures $T(z)$ are calculated

using an energy balance scheme in conjunction with an implicit finite difference (Crank-Nicolson) method (Jordan, 1991). The radiative transfer of energy is simulated using the Snow, Ice, and Aerosol Radiative Model that includes the evolution of the ice effective grain size (Flanner and Zender, 2006; Flanner et al., 2007).

In CLM(v5), the updated fractional compaction rate due to overburden pressure is adapted from Vionnet et al. (2012), expressed as

$$\left(\frac{1}{\Delta z}\frac{\partial \Delta z}{\partial t}\right)_{\mathrm{op}} = \frac{-c_\eta \sigma}{f_1 f_2 \eta_0 \rho \exp\left[a_\eta(T_f - T) + b_\eta \rho\right]}, \tag{4}$$


for each vertical layer with thickness $\Delta z$, constants $a_\eta = 0.1$ K$^{-1}$, $b_\eta = 0.023$ m$^3$kg$^{-1}$, a viscosity coefficient $\eta_0 = 7.62 \times 10^6$ kg s$^{-1}$ m$^{-2}$, and a tunable coefficient $c_\eta/(f_1 f_2)$ that depends on the liquid water content. van Kampenhout et al. (2017) also added a wind speed dependence to the initial snow density function in CLM that helps improve snow densities at the surface of ice sheets. To further improve ice sheet surface (and near-surface) densities, van Kampenhout et al. (2017) also added a

densification term to the compaction model that incorporates compaction due to drifting snow. They also increased the density





threshold $\rho_{\mathrm{dm}}$ (from 100 to 175 kg m$^{-3}$), from eq. (2), above which densification due to destructive metamorphism tapers off. By updating the CLM accordingly, van Kampenhout et al. (2017) eliminated a bias in the depth where the bulk density is equal to 550 kg m$^{-3}$, which was previously much too shallow. This *characteristic depth* is commonly used in firn studies as a single valued proxy for a given site's full density profile and also represents the transition from the first to the second stages of densification (Herron and Langway, 1980).

## 2.2 Firn densification models

Empirical firn densification models typically employ analytic functions that assume a steady-state density profile. They commonly define a critical density (usually 550 kg m$^{-3}$) above and below which represent two distinct stages of densification. Herron and Langway (1980), for example, demonstrate how their empirical firn densification model can predict observed density-depth relationships for the first two stages of densification given the mean annual temperature, annual accumulation rate, and surface density. When the climate is not changing, mean annual temperatures and accumulation rates are constant, eventually leading to firn well-approximated by the steady-state condition. Assuming a steady-state, a small parcel of firn has a vertical velocity $w$ relative to the surface, such that

$$w(z) = \frac{A}{\rho(z)},$$ (5)

where $A$ is the accumulation rate (in terms of SWE per year) and $\rho$ is the bulk density of firn at a given depth $z$ (Bader, 1954). Neglecting wind shear, the one-dimensional (kinematic) densification rate can then be derived from the material derivative

$$\frac{D}{Dt}\rho(z,t) = \frac{\partial\rho}{\partial z}w(z,t) + \frac{\partial\rho}{\partial t},$$ (6)

with $\partial\rho/\partial t = 0$ in a steady-state. Substituting eq. (5) into eq. (6) then gives the advective densification rate, which is closely related to the volumetric strain rate $\dot{\epsilon}$ via

$$-\frac{1}{\rho}\frac{D}{Dt}\rho(z) = -\frac{A}{\rho(z)^2}\frac{d\rho}{dz} = \frac{dw}{dz} = \dot{\epsilon}.$$ (7)

This steady-state approach is useful for deriving realistic density profiles and vertical strain rates in dry snow zones, but does not provide a dynamic representation of physical processes simulated in modern ESMs.

A dynamic, numerical densification model integrates fractional compaction rates for each snow element on a multi-layer, vertical grid. For example, Arthern et al. (2010) and Ligtenberg et al. (2011) developed a semi-empirical model based on measured firn thinning rates that can be coupled to the heat equation to calculate time dependent densification rates. Accordingly, the densification rate $\partial\rho/\partial t$ can be expressed in terms of temperature $T$, bulk density $\rho$, overburden pressure $P$, and an effective snow grain radius $r_e$, such that

$$\frac{1}{\rho}\frac{\partial\rho}{\partial t} = \frac{k_c \exp\left(\frac{-E_c}{RT}\right)\left[\frac{\rho_i}{\rho} - 1\right]P}{r_e^2},$$ (8)

with activation energy $E_c$ (60 kJ mol$^{-1}$), universal gas constant $R$ (8.31 J K$^{-1}$ mol$^{-1}$), ice density $\rho_i$ (917 kg m$^{-3}$), and $k_c = 9.2 \times 10^{-9}$ kg$^{-1}$m$^3$s for $\rho \leq 550$ kg m$^{-3}$ or $k_c = 3.7 \times 10^{-9}$ kg$^{-1}$m$^3$s for $\rho > 550$ kg m$^{-3}$. Adjustment of the rate





coefficient $k_c$ for $\rho \leq 550$ kg m$^{-3}$ is necessary to capture greater densification rates during the first stage of densification due to grain-boundary sliding (Alley, 1987). Because their study was confined to sites on or near the relatively warm Antarctic Peninsula, however, the model should not be applied in ESMs to represent the cold interior regions of Greenland and Antarctica without further calibration.

## 3 Methods

### 3.1 Firn model development

In this section, we describe model development resulting in a set of ELM configurations used to simulate the accumulation of snow and firn densification on land ice. Our modifications in E3SM began with the snow model in ELM(v1), which was inherited from the CLM(v4.5) used in CESM(v1). After adopting the 12 layer grid introduced into CLM(v5) by van Kampenhout et al. (2017), we expanded and modified their improved vertical grid to improve the spatial resolution at snow depths of 10 to 60 m. This improved spatial resolution is necessary to better simulate relevant processes at depths (and overburden pressures) more typical of firn.

#### 3.1.1 Expanded layering scheme

Guided by firn model improvements in CLM(v5) and used in CESM(v2) (van Kampenhout et al., 2017; Lawrence et al., 2019; Muntjewerf et al., 2020), we increased the maximum number of snow layers from 5 to 12 and eventually to 16 (Fig. 1). The upper-most nine layers vary in their minimum and maximum allotted thicknesses and conform to the top nine layers used in the 12 layer grid described by van Kampenhout et al. (2017). Expanding their 12 layer grid to a new 16 layer grid, we modified layers 10 and 11 (counting from the top) and appended onto the bottom an additional 4 so that layers 10–15 have an equal (7.67 m) spacing. When the snowpack becomes deep enough to fill these 15 layers, a semi-infinite (bottom-most) layer is created. This new layering scheme yields a vertical resolution of roughly 8 m for firn depths down to 60 m while maintaining its semi-infinite capability. It also maintains the variable spacing near the snowpack surface that is needed to resolve high temperature gradients and simulate radiative transfer.

Equipped with a 16 layer vertical snowpack grid, the simulation of firn in the ELM is enabled by removing the upper bound on the mass of snow (1 m SWE). This allows for the growth of semi-infinite firn columns as snow accumulates. With these changes in place, a baseline ELM configuration (hereafter referred to as "A'76," short for Anderson (1976)) is used to evaluate the original snowpack and firn densification scheme. For dry snow-zones (columns without any melting), vertically dependent, fractional compaction rates in this configuration are calculated as the sum of eq. (2) and eq. (3).

#### 3.1.2 Improved ice sheet surface densities

In the A'76 configuration (Sect. 3.1.1), snowfall that accumulates during a time step is assigned a density independent of the wind speed. This temperature-only dependence is one of the model deficiencies attributed to ice sheet surface densities that




are too low. To address this and other missing physics, two key improvements from van Kampenhout et al. (2017) were implemented into ELM. These include a wind speed dependence in the fresh snow density parameterization and the addition of a densification term that incorporates compaction (up to 350 kg m$^{-3}$) due to drifting snow. For the purpose of comparing experimental results to the current CLM(v5) configuration, we updated the ELM accordingly, including the overburden com-150 paction parameterization from Vionnet et al. (2012), shown in eq. (4). These updates, combined with CLM's current 12 layer grid as well as a destructive metamorphism (dm) density threshold $\rho_{\mathrm{dm}}$ = 175 kg m$^{-3}$ in eq. (2), are fully described by van Kampenhout et al. (2017), which we refer to here as "vK'17."

### 3.1.3 Compaction due to snowpack load pressure

Next, compaction due to the overburden pressure from snow loading was updated according to Arthern et al. (2010) by replac-155 ing eq. (3) with eq. (8). Here, the overburden pressure (or "load") is represented in the column as a function of the overlying, integrated column density $\sigma$, so that

$$P = g\left(\frac{\rho_i}{\rho}\right)\sigma, \tag{9}$$

where $g$ is the acceleration due to gravity (9.81 m s$^{-2}$) and the factor $\rho_i/\rho$ approximates the increased grain-load stress due to pore space within the snowpack (Cuffey and Paterson, 2010). From the "vK'17" configuration (Sect. 3.1.2), another 160 configuration, hereafter referred to as "A'10" (short for Arthern et al. (2010)), is defined using this updated (semi-empirical) overburden pressure model, which also includes the major improvements from van Kampenhout et al. (2017), described in Sect. 3.1.2. In this configuration, we keep the density threshold $\rho_{\mathrm{dm}}$ in eq. (2) at 175 kg m$^{-3}$ and apply our 16 layer grid (Fig. 1).

Later (but not included in A'10), we added a fresh snow compaction term, calibrated for dendritic snow, from (Lehning et al., 165 2002), where

$$\left(\frac{1}{\Delta z}\frac{\partial \Delta z}{\partial t}\right)_{\mathrm{dendritic}} = \frac{-g\sigma}{0.007\rho^{4.75-\frac{T_c}{40}}}. \tag{10}$$

Because ELM does not specify snow grain shape or type, we added this term only for snow having a low-enough snow grain size (i.e., where its layer-dependent optical sphere equivalent radius is less than roughly 80 μm) to be considered dendritic.

### 3.2 Statistical modeling

Implementation of candidate firn densification models into ELM provides a direct method of evaluation against observations. Although such comparisons provide complete tests, they require century scale simulations that, because of their computational expense, hinder the outcome of sensitivity studies needed for development. To circumvent this problem, statistical modeling was used to estimate empirical compaction rates, which are then used to calibrate the more physically-based models discussed above (A'76, vK'17, and A'10).

To approximate a statistical model of the dry snow zones across Antartica and Greenland, Monte Carlo experiments were conducted to generate thousands of plausible firn density-versus-depth profiles. First, random mean-annual temperatures ($T$)





below -25 °C were drawn from the left tail of a Gaussian distribution (with an estimated global mean $\mu$ = 14.9 °C and standard deviation = 16 °C) selected to give a distribution of temperatures crudely representative of Earth's land surface. Next, accumulation rates ($A$) were drawn at random from a lognormal distribution selected to give values representative of warm

($T$ > -51 °C) or cold ($T \leq$ -51 °C) dry snow zones, with $0.07 < A < 0.4$ or $A < 0.07$ m SWE yr$^{-1}$, respectively. Valid mean annual temperature and accumulation rate pairs were then combined with a surface density (ranging from 300 to 380 kg m$^{-3}$) drawn at random from a uniform distribution. These values were inserted in the empirical model of Herron and Langway (1980) from which we derive empirical steady-state strain rates (with a vertical resolution of 10 cm) from eq. (7).

From our estimated empirical strain rate-versus-depth data, we optimized the previously described densification model

coefficients (from A'76, vK'17, and A'10) by applying a regularized least squares algorithm for two stages of densification (above and below $\rho = 550$ kg m$^{-3}$). Normalized covariances (i.e., correlation R) between empirical strain rates and those from optimized firn densification parameters were inferred to evaluate the skill of each set of parameters.

### 3.3 Firn density simulations

Although statistical modeling enables a basic evaluation of experimental densification models, it does so using empirically

derived, steady-state density profiles, rather than those derived from the densification models themselves. To calculate density profiles that are truly stable for a given densification model, a numerical scheme is needed to integrate accumulating firn to a steady state. The Common Infrastructure for Modeling the Earth (CIME) provides the software necessary to fully simulate the accumulation of snow and densification of firn on land ice with ELM. In ELM stand-alone mode, the surface boundary condition is provided by atmospheric re-analysis data that include diurnally (3 hour) varying precipitation, solar radiation,

temperature, and wind speed from the Climate Research Unit and the National Center for Environmental Prediction (CRUN-CEP) (Viovy, 2018). The following historical climate simulations were conducted using coarse-resolution, stand alone ELM (an "I-compset" at ne11 resolution).

#### 3.3.1 Equilibrium climate

To spin-up the model, we initialized (starting on January 1st) snowpack conditions using a thin (50 mm SWE) cover on ELM's

glacier columns (including those representing the Greenland and Antarctic Ice sheets) as well as all land columns located north of 44 °N and simulated 260 years of snow accumulation and densification under a repeating (1901-1920) atmospheric forcing. This initial, *cold start*, condition removes any prior assumption about the natural firn density profiles, resulting in firn densities representative of densification processes simulated entirely within the model.

#### 3.3.2 Twentieth century climate

Twentieth century simulations are initialized using conditions at the end of 260-year equilibrium climate simulations (i.e., as "restart runs"). One-hundred years of firn densification over land ice were then simulated using atmospheric forcing data from





**Table 1.** Dry firn simulation attributes tested in the ELM. Experiments summarized here all include densification due to destructive metamorphism, as shown in eq. (2), but with different density thresholds $\rho_{\mathrm{dm}}$ (footnotes $a, b, c$). Modifications to original parameterizations are indicated below.

| Experiment ID | Max. layers | Fresh snow density | Snowdrift | Overburden compaction |
|---|---|---|---|---|
| A'76[a] | 16 | Anderson (1976) | None | Anderson (1976) |
| vK'17[b] | 12 | van Kampenhout et al. (2017) | van Kampenhout et al. (2017) | Vionnet et al. (2012) |
| A'10[b] | 16 | van Kampenhout et al. (2017) | van Kampenhout et al. (2017) | Arthern et al. (2010) |
| vK'17+[c] | 16 | van Kampenhout et al. (2017) | van Kampenhout et al. (2017) | Vionnet et al. (2012)[def] |

[a] $\rho_{\mathrm{dm}} = 100$ kg m$^{-3}$, [b] $\rho_{\mathrm{dm}} = 175$ kg m$^{-3}$, [c] $\rho_{\mathrm{dm}} = 150$ kg m$^{-3}$, [d] $f_2 = 4.9$, [e] added constant (-1.18 × 10$^{-10}$ s$^{-1}$), [f] added eq. (10) for dendritic snow (Lehning et al., 2002)

1901-2000 (Viovy, 2018). In total, this combined procedure gives 360 years of snow accumulation and densification ending in the year 2000, enabling direct comparisons with recent firn density measurements.

Firn density measurements were collected by Mosley-Thompson et al. (2001), in Greenland, and Lamorey (2003), in West
Antarctica (Siple Dome). The measurements from Greenland include roughly 1 m sectioned cores from 31 sites that each extend at least 9 m below the surface. A core from North Dye (2BA) extends 120 m below the surface with densities ranging from 341 to 917 kg m$^{-3}$. The Siple Dome cores were also cut into 1 m sections and weighed in the field. Densities from seven cores extending at least 29 m below the surface, including four beyond 91 m, ranged from 415 to 914 kg m$^{-3}$. To compare these measurements to ELM simulation results, which represent relatively large grid cells, each set of cores are averaged into
a composite profile; one representing the GrIS and one representing Siple Dome.

## 4 Results and Discussion

To evaluate firn density simulations, we test a series of compaction parameterizations in ELM with a semi-infinite vertical snowpack grid (Table 1). We organize equilibrium climate simulation results by various mean annual temperatures and show data only for accumulations representative of dry-snow zones across the Greenland and Antarctic Ice Sheets (i.e., less than
0.5 m SWE yr.$^{-1}$). To improve the accuracy of our firn model simulations, we optimize compaction terms against empirical strain rates using statistical modeling. Finally, we compare our simulation results to the recent (1998-2000 C.E.) firn density measurements by Lamorey (2003) and Mosley-Thompson et al. (2001).

### 4.1 Evaluation of steady-state density profiles

#### 4.1.1 ELM equilibrium climate simulations

We start with a baseline configuration (A'76) forced with atmospheric reanalyses repeated over the years 1901-1920. After 13 repeated cycles lasting a total of 260 years, we calculate steady-state density profiles averaged from the final 100 years of





simulation results (Fig 2.). In these initial simulations, densities are highly variable in the top 3 m. Densities increase from less than 300 kg m$^{-3}$ at the surface to as much as 600 kg m$^{-3}$ within the first 5 m for relatively warm dry snow zones where the mean annual temperature is within a couple degrees of -25 °C. Compared to the widely used steady-state empirical model

from Herron and Langway (1980), ELM-simulated densities become too large below the top two meters. High densities persist down to a depth of 20 m for all temperature regimes, a model deficiency also identified and described in van Kampenhout et al. (2017).

To improve firn density simulations, we implemented key changes from van Kampenhout et al. (2017) that result in more accurate surface densities and changed the overburden pressure compaction parameterization to the semi-empirical model pro-

posed by Arthern et al. (2010). We initialized snowpack conditions and simulated firn densities in the same manner described previously. These simulations demonstrate a stronger effect of temperature on densification rates, resulting in more variation in density with depth (Fig 3). For regions where the mean annual temperature is greater than -32 °C, this dynamic implementation of eq. (8) in the ELM results in characteristic depths (where $\rho = 550$ kg m$^{-3}$) more consistent with Herron and Langway (1980) than when using the original compaction parameterization, from eq. (3). For regions where the mean annual temperature

is greater than -25 °C, near surface densities vary considerably, ranging from below 550 to as much as 830 kg m$^{-3}$, indicating the formation of ice lenses within the upper most 5 meters. In the original model (A'76), glacier surface densities are as low as 150 kg m$^{-3}$, while those simulated using a better fresh snow density parameterization and an added wind packing term (van Kampenhout et al., 2017) result in surface densities closer to 300 kg m$^{-3}$.

To better understand what drives rapid densification near the surface, we approximate vertical strain rates (negative for

deformation) and firn age from Sorge's Law (Bader, 1954). Assuming a steady-state, we compare densities and fractional compaction rates as a function of firn age (Fig. 4). These data show rapid densification within the first ten years. Near the surface, fractional compaction rates, which are equivalent to negative strain rates, can become as large as 10$^{-6}$ s$^{-1}$ (9 % per day), after which they decrease by three orders of magnitude (to below 3 % per year) after just 20 years. Densification tapers-off at lower densities (around 450 kg m$^{-3}$) for colder climates, a temperature-dependent effect enhanced with the model from

Arthern et al. (2010).

To test our statistical calibration method, we repeated ELM experiments, but replaced the pressure compaction equation with an optimized version described by Vionnet et al. (2012). We also set the density threshold $\rho_{\mathrm{dm}}$ in eq. (2) to 150 kg m$^{-3}$, reverting back to its original value given by Anderson (1976). This optimized configuration produces compaction rates at least moderately correlated (R$^2$ = 0.4) with empirical modeling, though only for the second stage of densification ($\rho >$550 kg m$^{-3}$).

For reference, we compare steady-state density profiles from this additional ELM experiment, referred to as "vK'17+," to the A'10 ELM experiment (Fig. 3 and Fig. 4). Note also (from Table 1) that this vK'17+ configuration includes the dendritic snow compaction term from eq. (10) (Lehning et al., 2002), though this addition did not have a noticeable effect on the density profile below 1 m. Like in the original CLM (i.e., our vK'17 run), results from vK'17+ show slightly improved top 10 m near-surface densities for warm regions, but under-estimate densification as depths increase.

Although the semi-empirical model improves the density profile, its implementation into ELM, does not go far enough to represent the main drivers of snow compaction that occur at relatively low densities ($<$ 500 kg m$^{-3}$). With simple represen-





tations of compaction due to destructive metamorphism and other relevant processes (e.g., grain-boundary sliding), modeling challenges near the surface arise where densities range from 300 to 500 kg m$^{-3}$ and vary due to sub-grid scale snow microstructure properties not yet accounted for in firn models (Lundin et al., 2017). Despite widespread over-densification of

near-surface firn, colder regions develop steady-state density profiles that vary too weakly with depth. Their effective strain rates, however, are similar to those predicted with empirical modeling for low accumulation rates (i.e., those less than 0.15 m SWE yr$^{-1}$). These results indicate that the 1901–1920 atmospheric forcing data contain accumulation rates (from snowfall) that are too high in colder regions of the ice sheet. Unrealistically high accumulation rates, and the resulting faster downward advection of firn relative to the surface, explains why kinematic densities (density-versus-depth relationships) appear too low

while undergoing realistic compaction rates.

At this point, two ELM configurations, labeled vK'17 and vK'17+, are nearly identical, but they have two important distinctions: different layering schemes (12 versus 16) and different model constants. After calibration, our new (vK'17+) implementation is given by:

1. decreasing the destructive metamorphism coefficient $c_3$ (Anderson, 1976; van Kampenhout et al., 2017) from 2.777 $\times$
$10^{-6}$ to 0.83 $\times$ $10^{-6}$ s$^{-1}$;

2. increasing the overburden pressure compaction viscosity correction factor $f_2$ (Vionnet et al., 2012) from 4.0 to 4.9;

3. adding a constant compaction term equal to (-)1.18 $\times$ $10^{-10}$ s$^{-1}$.

As expected, these changes, representing an a priori calibration, better capture densification below 10 m, slightly improving the steady-state comparison between our simulations and Herron and Langway (1980). Because our a priori calibration uses a

mean annual temperature domain that includes the coldest regions of the Antarctic ice sheet, we might expect further improved density profiles for warm regions by restricting our calibration to mean annual temperatures greater than -34°C. Also worth mentioning is that similar results can be produced using the original pressure compaction term from Anderson (1976) when increasing the viscosity coefficient $\eta_0$ by factor of 50. This is because the viscosity $\eta$ has essentially the same functional form as the one from Vionnet et al. (2012), but with slightly different coefficients (for which we adjust via a least squares regression).

**4.1.2   Regression modeling**

To improve model accuracy, we use Monte Carlo experiments to calibrate densification model coefficients. Covariance matrices, indicating model specific a priori variances and inter-model covariances (and correlations), can be generated for multiple experimental parameterizations without conducting full ELM simulations (Table 2). Diagonal entries of the multi-model covariance matrix give model-specific variances (SD$^2$), while off-diagonal entries give covariances that are normalized to calculate

empirical correlations (R). Surprisingly, a slightly modified destructive metamorphism expression from eq. (2) (Anderson, 1976), by itself, results in compaction rates that are more highly correlated with the empirical model (Herron and Langway, 1980) than that from unmodified, combined parameterizations that include both destructive metamorphism and compaction due to overburden pressure. We found this higher correlation by changing the density tapering constant (from 46 to 12.5 cm$^3$g$^{-1}$)





**Table 2.** Steady-state firn strain rate regression modeling. Models' root mean squared error (RMSE) and coefficient of determination ($R^2$) are estimated from training data compared against a two-stage empirical model. The models' two stages of densification each have different variabilities as indicated by their standard deviations (SD).

First stage (300 – 550 kg m$^{-3}$)

| Densification model | RMSE ($\times 10^{-10}$ s$^{-1}$) | $R^2$ | SD ($\times 10^{-10}$ s$^{-1}$) |
|---|---|---|---|
| *Empirical* (Herron and Langway, 1980) | 0 | 1 | 2.5 |
| Semi-empirical + metamorphism (Arthern et al., 2010; Anderson, 1976) | 1.9 | 0.50 | 2.3 |
| Metamorphism (Anderson, 1976)[a] | 2.3 | 0.13 | 0.90 |
| Pressure + metamorphism (Anderson, 1976) | 2.3 | 0.14 | 0.92 |
| Pressure + metamorphism (Vionnet et al., 2012; Anderson, 1976) | 2.3 | 0.13 | 0.91 |

Second stage (550 – 800 kg m$^{-3}$)

| Densification model | RMSE ($\times 10^{-10}$ s$^{-1}$) | $R^2$ | SD ($\times 10^{-10}$ s$^{-1}$) |
|---|---|---|---|
| *Empirical* (Herron and Langway, 1980) | 0 | 1 | 0.4 |
| Semi-empirical + metamorphism (Arthern et al., 2010; Anderson, 1976) | 0.30 | 0.66 | 0.35 |
| Metamorphism (Anderson, 1976)[a] | 0.28 | 0.50 | 0.28 |
| Pressure + metamorphism (Anderson, 1976) | 0.30 | 0.42 | 0.26 |
| Pressure + metamorphism (Vionnet et al., 2012; Anderson, 1976) | 0.30 | 0.43 | 0.26 |

[a] modified destructive metamorphism expression in eq. (5) ($c_5 = 12.5$ cm$^3$g$^{-1}$)

in eq. (2). This modified destructive metamorphism expression is most highly correlated with empirical compaction rates for

densities greater than 600 kg m$^{-3}$, but does not co-vary for densities less than 450 kg m$^{-3}$. Given that its original purpose was to model low density (less than 250 kg m$^{-3}$) compaction rates undergoing temperature dependent metamorphism, its moderately high correlation with empirical strain rates at relatively high densities does not justify a sufficiently quantitative densification mechanism.

    Using the semi-empirical, pressure-driven model developed by Arthern et al. (2010), we are able to capture more variability

($R^2 = 0.67$) in strain rates in the colder regions across the interior of the Greenland and Antarctic Ice Sheets (Fig 5). This result is encouraging, as this semi-empirical model is constructed with data from the Antarctic Peninsula and Berkner Island.

    After applying the method of least squares for two stages of densification, the optimized models have smaller variances than the empirical model. A lower model variance occurs when it does not covary with the empirical model. This effectively reduces a model's prediction risk if it does not also result in an increased bias. Although further statistical inference can assess this

bias-variance tradeoff, we can directly evaluate model performance with ELM simulations.

    Across the ice sheets' dry snow zone temperature domain, roughly -60 to -25 °C, these results indicate negative correlations between overburden pressure and empirical strain rates. Consequently, regression models, which we use to estimate model





coefficients, minimize pressure compaction terms to calibrate for decreasing strain rates with increasing overburden pressure. Without including either a destructive metamorphism term or grain-size dependence in the model, the regression coefficient

for a pressure term can even be negative. While firn densification is usually considered analogous to *pressure sintering*, where porous materials deform under stress, Meussen et al. (1999) determined that the density itself is the most crucial variable in approximating the densification rate. These findings, as well as other firn model inter-comparison studies (Lundin et al., 2017; Verjans et al., 2019), emphasize the need for further development of constitutive, density-dependent stress versus strain relationships which can also account for microphysical, grain-scale features.

While the semi-empirical firn densification model is tailored for Antarctic Peninsula sites, we hypothesize its implementation in ELM will improve density profiles on a site-by-site basis after optimizing its coefficients from temperatures and accumulation rates that are representative of its target domain, including the ice sheet interior. Even without optimization, we find improvements in the simulated density profile in equilibrium simulations, though our analysis with ELM thus far is limited to a generalized comparison with a broad (climate) perspective rather than to a more site-specific comparison against

direct observations. Although these developments are ongoing, we might expect improved near-surface densities, based on our statistical computing, with the following changes to the A'10 experiment:

1. To further improve surface densities, add the expression for dendritic snow compaction from eq. (10) (Lehning et al., 2002), as we did for vK'17+;

2. Set the destructive metamorphism coefficient $c_3 = 5.8 \times 10^{-7}$ s$^{-1}$, from eq. (2);

3. Set the creep coefficient $k_c = 1.4 \times 10^{-9}$ kg$^{-1}$m$^3$s when $\rho \leq 550$ kg m$^{-3}$ and $k_c = 1.2 \times 10^{-9}$ kg$^{-1}$m$^3$s when $\rho > 550$ kg m$^{-3}$, from eq. (8);

4. Add a constant compaction term, as we did for vK'17+, but equal to (-)$2.02 \times 10^{-10}$ s$^{-1}$ when $\rho \leq 550$ kg m$^{-3}$ and equal to (-)$2.7 \times 10^{-11}$ s$^{-1}$ when $\rho > 550$ kg m$^{-3}$.

### 4.2    End of 20th century Greenland and Antarctic Ice Sheet case studies

To evaluate the ELM snow and firn densification mechanics against recent observations, we continued the steady-state experiments under a 1901–2000 atmospheric forcing. Using the semi-empirical formulation (A'10), eq. (8), as well as an optimized version of the current CLM (vK'17+), we simulated 100 additional years of accumulation and compaction. We compare our simulated end-of-century density profiles, averaged across the the GrIS and for grid-cells nearest to Siple Dome, Antarctica, to recent firn core measurements (Fig. 6). We average data in this manner (i.e., over the whole ice-sheet for Greenland versus

only 4 grid cells nearest Siple Dome) to most closely represent model results with respect to observations. In general, the semi-empirical model results in better simulated densities deeper than 20 m, while our optimized vK'17+ configuration captures near surface densities better. While variability can be large, particularly across the GrIS, both models show improvement compared to their original counterparts (ELM v1 and CLM).





Our ELM simulations are currently limited by their spatial resolution. Because calculations across spatially varying grid cells
are independent, i.e., they have no ability to influence one another, we seek a balance between computational expense versus
adequate spatial sampling. The coarse ne11 global resolution enables short wall-clock time, multi-century duration simulations
while still providing multiple grid cells on both ice sheets that develop deep ($> 60$ m) firn columns. These coarse-resolution
simulations do, however, present challenges in evaluating regions near coasts as atmospheric forcing data are averaged over
large grid-cells that include both land and ocean surface types. Near Siple Dome, Antarctica, for example, this large grid-
cell remapping lead to a cold bias, resulting in too-slow densification. Therefore, we adjusted our Siple Dome comparisons
to include grid-cells away from the coast that better represent atmospheric conditions and result in a more realistic density
simulation.

Encouragingly, our simulation results compare well with firn density measurements and indicate an improved capability in
the ELM. Moving forward, we have yet to address whether the new ELM firn model satisfactorily represents ice sheet SMB.
To address this question, we should focus on the near surface layers, as they contain the primary SMB components. To capture
melt water percolation, retention, and refreezing, it could be necessary to model the upper most 20 m, but accurate modeling
of firn densities all the way down to pore close-off might be unnecessary. As a result, the optimized version (vK'17+) is likely
the better choice for implementation into the next major release of the E3SM. Although it shows a low density bias at depth,
this bias might actually be preferable to a high bias if we aim to partition run-off regimes – which are characterized by thick,
dense, near-surface ice lenses – from percolation regimes.

On the other hand, we have yet to test in ELM an optimized version of the semi-empirical model, which could be pro-
gressively tailored to accurately reproduce near surface densities. Furthermore, instead of using Monte Carlo integration to
determine a theoretical mean annual temperature-accumulation probability distribution, an improved method would use data
from reanalyses or regional climate modeling as inputs to the empirical model from Herron and Langway (1980). Such an
approach could be tailored to dry-snow zones of the GrIS, for example, by restricting data to regions where the surface tem-
perature never exceeds 0°C.

The next major step, however, is a full evaluation of the model's firn densification mechanics that include the effects of
liquid water. Such an evaluation remains challenging, as observations in these regions are limited. Such limitations call for
more observational studies where the ice sheet is beginning to melt. In the near future, this could be the entire GrIS, so accurate
modeling of these regimes will be crucial.

## 5 Conclusions

We considered different compaction models to improve firn simulation in ELM, and compared simulated density profiles with
density measurements of firn cores. Expanding this evaluation to include regions across the dry snow zones of the Greenland
and Antarctic Ice Sheets, we examined the variation of density with depth in equilibrium climate simulations and compared
results with steady-state empirical models (Herron and Langway, 1980). Our analysis, similar to that by van Kampenhout
et al. (2017) for CLM, demonstrates the need for a better representation of firn (i.e., multi-year snow). As first shown by van





Kampenhout et al. (2017), our simulations suggest that the original snow overburden pressure compaction parameterization (Anderson, 1976) results in near-surface firn densities that are too large. This high bias for near surface densities also exists in our simulations with a semi-empirical densification model (Arthern et al., 2010). With guidance from van Kampenhout et al.

(2017), Vionnet et al. (2012), Anderson (1976), and Lehning et al. (2002), we improved simulated density versus depth relationships in dry snow zones, a first step toward implementing a valid dynamic firn densification routine within the E3SM Project (2018). With an evaluation of the simulation of dry firn densification, we have optimized the ELM firn model for future studies of the impacts of liquid water on firn density and SMB.

Ultimately, this study seeks to enable better predictions of SLR as a direct result of surface melt and mass loss from the

GrIS. Accurate partitioning of the SMB terms, however, remains a challenge in ESMs. Although regional climate models currently enable a higher spatial resolution, limitations in our understanding of supra-glacial hydrology still remain a source of uncertainty in determining Greenland's future contribution to SLR. Future developments will seek to improve the capability of simulating supra-glacial hydrology in the E3SM, including better quantifying surface melt, percolation, refreezing, and the build-up of perennial aquifers.

*Code and data availability.* The E3SM (version 1) source code is maintained and available at https://github.com/E3SM-Project. Source code modifications and model development that will reproduce these results are available from Edwards et al. (2020). ELM simulation data, associated python analysis scripts, and the offline statistical firn model referenced in this manuscript are archived and publicly accessible from Schneider (2020). The CLM (version 5) source code is maintained and available at https://github.com/ESCOMP/CTSM.

*Author contributions.* AS collected published data, modified E3SM source code, conducted ELM simulations and formal analyses, devel-

oped the statistical model, curated data, and prepared the figures, tables, and paper. CZ and SP supervised and co-led the project conceptualization, funding acquisition, and administration and helped write the paper.

*Competing interests.* The authors are not aware of any conflicts of interest regarding the publication of this manuscript.

*Acknowledgements.* This work is supported by the U.S. Department of Energy's Scientific Discovery Through Advanced Computing (LANL-520117) and Earth System Model Development (DE-SC0019278) programs. Source code was obtained from the Energy Exascale Earth

System Model project, sponsored by the U.S. Department of Energy, Office of Science, Office of Biological and Environmental Research. We thank all the scientists, software engineers, and administrators who contributed to the development of the Energy Exascale Earth System Model and the Community Earth System Model.

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



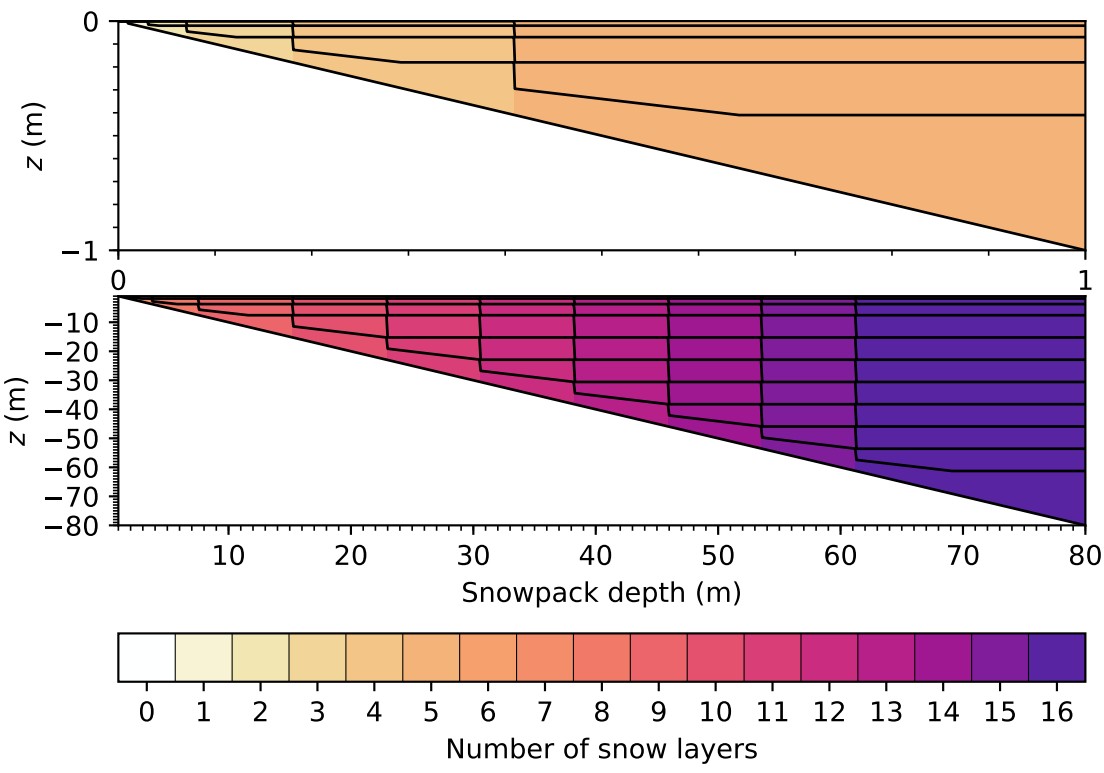

**Figure 1.** Original ELM (5 layer) vertical snowpack grid (top; 1 m) beneath which the new firn model appends up to 11 new layers that can extend as deep as 80 m. The dynamic grid can, based on the total snowpack depth, adjust every time-step the total number of layers (maximum of 16) and the bottom two layers' thicknesses.





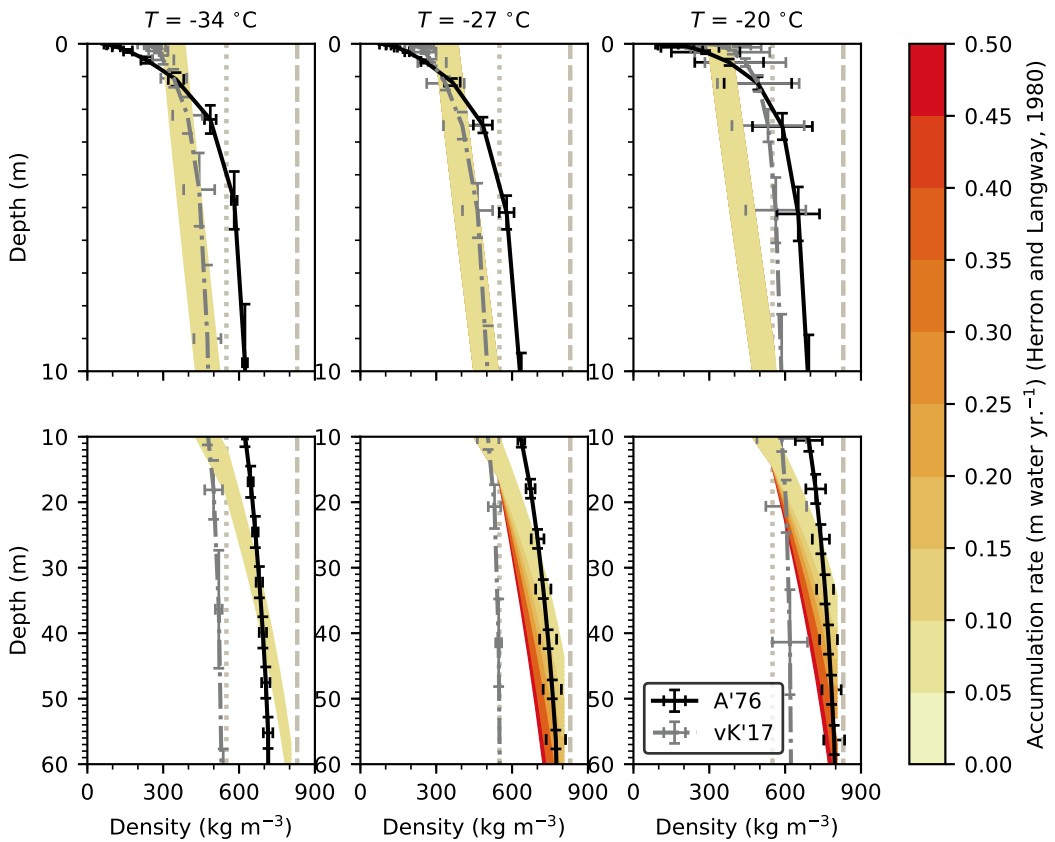

**Figure 2.** Steady-state density profiles for three mean annual surface temperatures (by column). Line graphs show 100-year means from ELM simulations (after expanding the original 5-layer snowpack grid). Shown in black are results using the original pressure compaction model, but with our 16-layer grid, while results in gray show simulation results using the (12-layer) CESM Land Model (CLMv5) scheme. Error bars represent grid-cell standard deviations. Vertical, light gray lines show fixed reference densities of firn at the characteristic firn depth (dotted, 550 kg m$^{-3}$) and pore close-off (dashed, 830 kg m$^{-3}$). Empirical modeling results, which indicate the target density-versus-depth relationships for a given mean annual temperature, are shown for various plausible accumulation rates (0.11–0.50 m SWE yr.$^{-1}$), as indicated by the colorbar, and for a range of surface densities (300–380 kg m$^{-3}$). Note vertical scale change between results in upper 10 m (top row) and 10–60 m (bottom row).





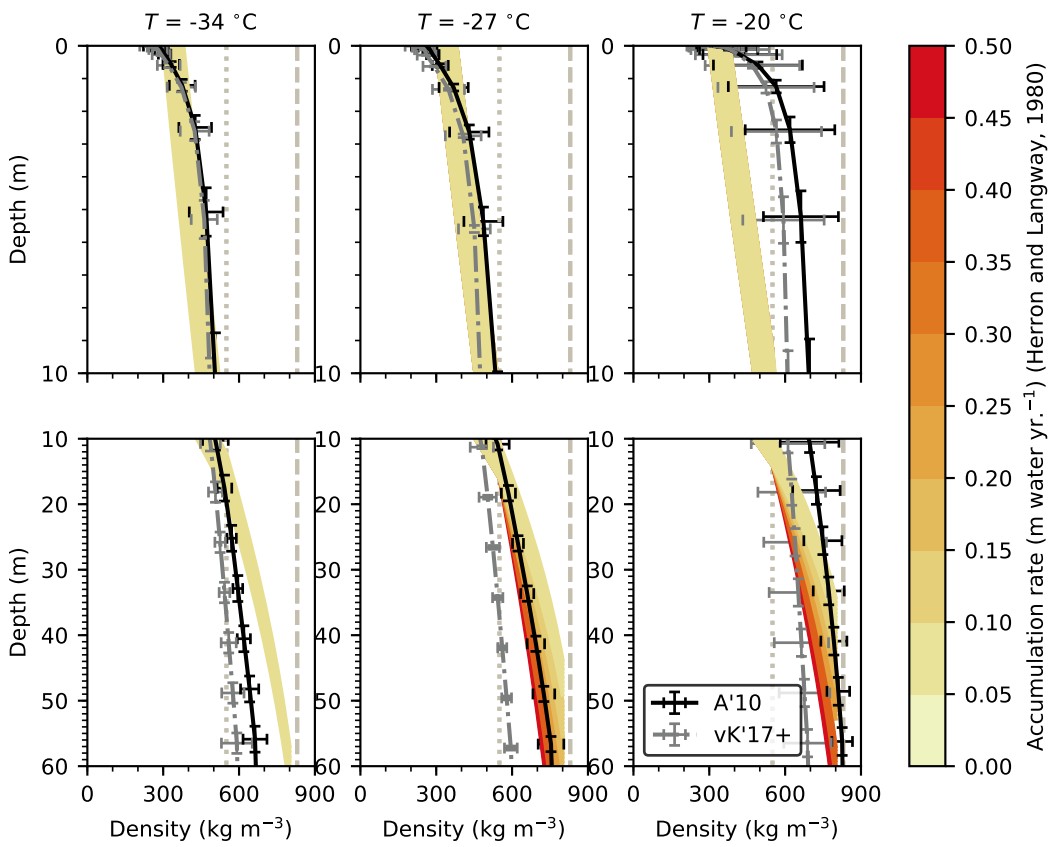

**Figure 3.** As in Fig. 2, with the "A'10" and "vK'17+" experiments (Table 1). Black line graphs show 100-year means from ELM simulations after expanding the snowpack grid, improving surface densities, and replacing the original pressure compaction model with a semi-empirical compaction model from Arthern et al. (2010). Results in gray show simulation results using an optimized, 16 layer version of the current CESM Land Model scheme. Error bars represent grid-cell standard deviations. Light gray lines show fixed reference densities of firn at characteristic firn depth (dotted, 550 kg m$^{-3}$) and pore close-off (dashed, 830 kg m$^{-3}$). Empirical modeling results, which indicate the target density-versus-depth relationships for a given mean annual temperature, are shown for various plausible accumulation rates (0.11–0.50 m SWE yr.$^{-1}$), as indicated by the colorbar, and for a range of surface densities (300–380 kg m$^{-3}$). Note vertical scale change between results in upper 10 m (top row) and 10–60 m (bottom row).



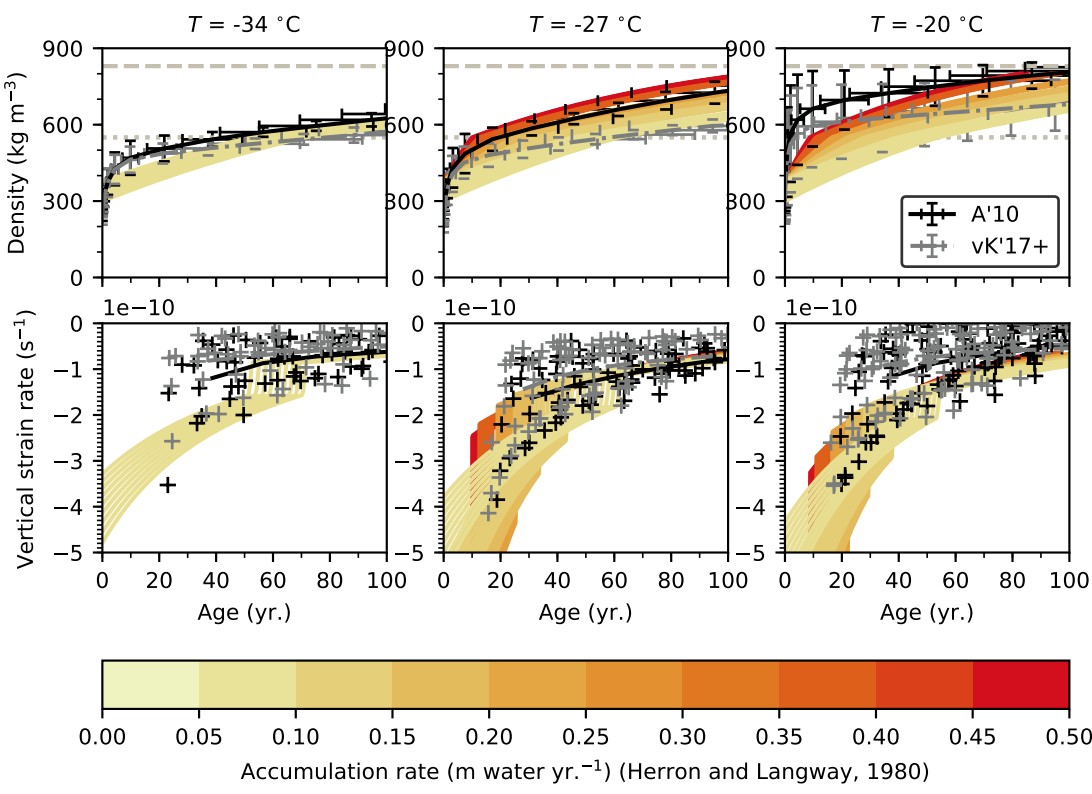

**Figure 4.** Steady-state densification, including densities (top) and advective strain rates (× 1e-10, bottom) as a function of firn age for various mean annual temperatures (by column). Line graphs show 100-year means from ELM simulations after expanding the snowpack grid to 16 layers. Shown in black are results using a semi-empirical pressure compaction model (A'10) from Arthern et al. (2010), while results in gray show simulation results using an optimized, 16 layer version of the current CESM Land Model scheme (vk'17+). Error bars represent grid-cell standard deviations. Empirical modeling results, which indicate target densities and strain rates for a given mean annual temperature, are shown for various plausible accumulation rates (0.11–0.50 m SWE yr.$^{-1}$), as indicated by the colorbar, and for a range of surface densities (300–380 kg m$^{-3}$).

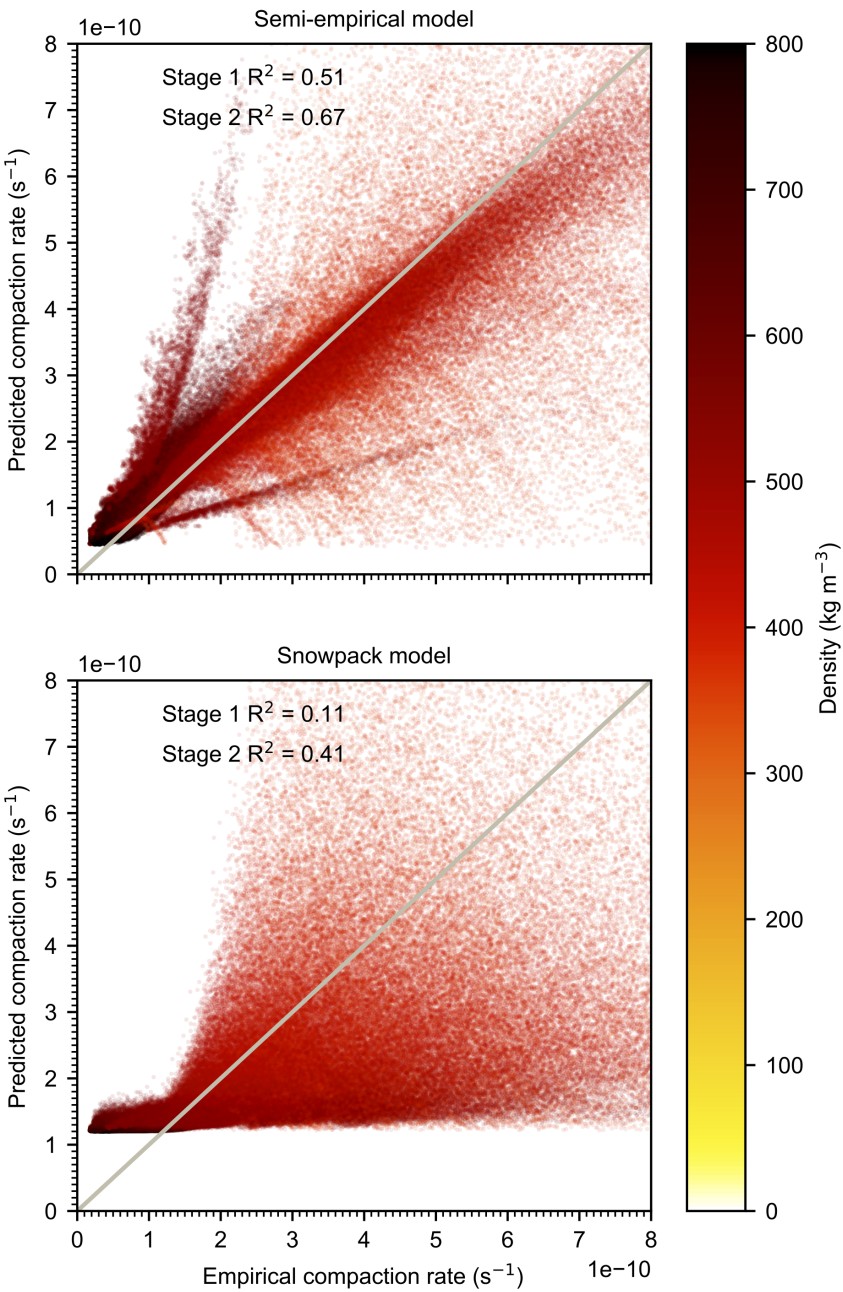

**Figure 5.** Firn model calibration for two stages of densification (1. $\rho < 550$ kg m$^{-3}$; 2. $\rho > 550$ kg m$^{-3}$). Empirical density profiles ($N = 1000$) are generated from Herron and Langway (1980) for a wide range of climate conditions representative of the Greenland and Antarctic ice sheet dry snow zones. Depth-dependent empirical compaction rates can be approximated from these steady-state profiles, enabling calibration of experimental densification models. The top panel shows predicted compaction rates from an optimized linear combination of destructive metamorphism (Anderson, 1976) and pressure-driven (Arthern et al., 2010) compaction terms. The bottom panel shows a similar combination, instead using the pressure-driven compaction model from Vionnet et al. (2012).



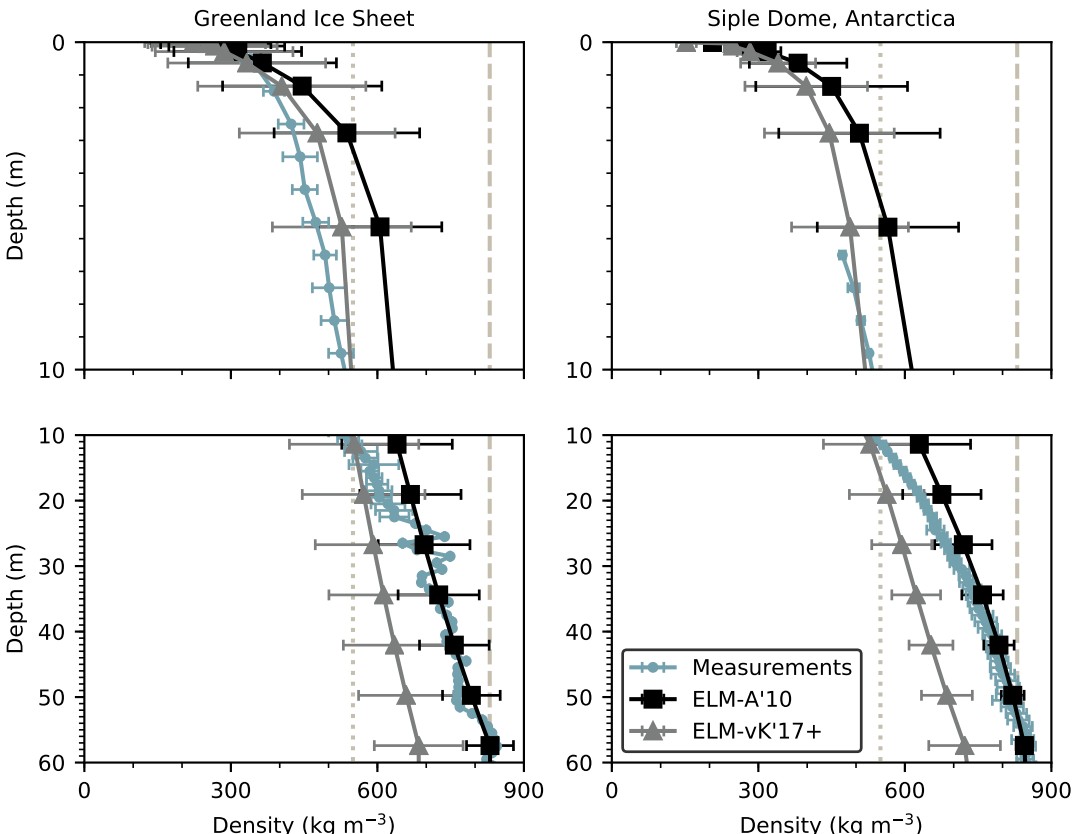

**Figure 6.** End of 20th century firn density profiles simulated with ELM compared to firn core measurements from Greenland (Mosley-Thompson et al., 2001) and Siple Dome (Lamorey, 2003). Line graphs represent ELM simulations with two different densification models: a semi-empirical formulation (Arthern et al., 2010), in black; and a calibrated, 16-layer version of the current CESM Land Model configuration, in gray. ELM simulation results from Greenland represent regional means calculated from 8 model grid cells with mean annual temperatures less than or equal to -25°C. ELM simulation results from Antartica represent regional means calculated from 4 grid-cells located near Siple Dome.