# Peer review of "Snowpack and firn densification in the Energy Exascale Earth System Model (E3SM) (version 1.2)"

_Geoscientific Model Development, 2020_

## Referee Comment (RC1) · Vincent Verjans (Referee) · 27 Aug 2020

This study explores the implementation of dry snow and firn densification formulations in the Land Model of the Energy Exascale Earth System Model. Firstly, the snowpack domain is extended to greater depths and simulated at a higher vertical resolution. Secondly, the authors compare the performance of four different densification formulations with empirical strain rates generated from the analytical firn model of Herron and Langway (1980). Finally, they evaluate results of two of the four formulations with some firn cores from the Greenland ice sheet and from Siple Dome in Antarctica.

Improving firn densification schemes in Earth System models is a valuable objective. However, many aspects of the study need to be significantly improved to possibly meet the standards of Geoscientific Model Development (GMD). I list below my concerns about the contents and the structure of the manuscript. I relate my remarks to some review criteria, which are available on the website of GMD. I also try to outline possible avenues for the authors to improve their work. My comments are separated in Major comments, which address general shortcomings of the study, and Specific comments.

Major Comments
1) The lack of information about the optimisation
Here are two review criteria of GMD:

     *Are the methods and assumptions valid and clearly outlined?*
     *Is the description sufficiently complete and precise to allow their reproduction by fellow scientists*
     *(traceability of results)?*

I focus first on the optimisation method applied (Section 3.2). The entire optimisation method is described in a single sentence (lines 184-186):
"*From our estimated empirical strain rate-versus-depth data, we optimized the previously described densification model coefficients (from A'76, vK'17, and A'10) by applying a regularized least squares algorithm for two stages of densification (above and below $\rho = 550$ kg $m^{-3}$).*"
The A'76 model includes 7 different coefficients ($c_1, c_2, c_3, c_4, c_5, \rho_{dm}, \eta_0$), the vK'17 includes 10 ($c_3, c_4, c_5, \rho_{dm}, \eta_0, c_\eta, f_1, f_2, a_\eta, b_\eta$) and the A'10 includes 7 ($c_3, c_4, c_5, \rho_{dm}, k_c^{\rho<550}, k_c^{\rho>550}, E_c$). Additionally, in the Results section, the authors mention "*adding a constant compaction term*" (line 277), which does not appear in any equation and is not explained in the Methods section. Throughout the manuscript, the authors never state which coefficients are subject to the optimisation. Moreover, they mention in Section 4.2 that " *we have yet to test in ELM an optimized version of the semi-empirical model*". From my understanding, the "*semi-empirical model*" is A'10 and they decided to compute the ELM simulations with the original version of A'10 and not the optimised version. In contrast, for some reason, the authors did the ELM simulations with the optimised version (vK'17+) of vK'17. They still provide speculative avenues for a re-parameterisation of A'10 at lines 322-328. These statements are not supported by any quantitative information about a better fit of the optimised A'10 either to observed data or to the strain rates generated from the model of Herron and Langway (1980) (referred to as HL hereafter). Finally, they assert that they optimise A'76 ("*we optimized the previously described densification model coefficients (from A'76, vK'17, and A'10)*"). However, the only information to be found about the modifications brought to the model is the change of the value of $c_5$, but nothing about other parameters and nothing about a performance comparison between the original A'76 and the optimised version.
Coming back to the optimisation methodology itself, the authors decide to select annual mean temperatures only below -25 °C, but they then proceed to model simulations for grid cells where the annual mean temperature is as high as -20 °C (Section 3.3, Figures 2, 3 and 4). It is puzzling that the authors themselves suggest a better approach to selecting mean annual temperatures and accumulation rates, which would be easy to implement (lines 357-361). They also claim to calibrate the models by matching the computed strain rates to the HL strain rates. The issue is that several models use dynamic variables in the strain rate equations: $T$, $\sigma$, $P$ and $r_e$ in Eqs 1, 2, 3, 4, 8, 9 and 10. The HL can provide analytic solutions of strain rates for steady state annual mean temperature and accumulation. The dynamic models require values for the dynamic variables at each time step and for each layer of the firn column. A steady state annual mean temperature does not correspond to all firn layers having the same temperature year-round (temperature still varies seasonally when in steady state). Similarly, accumulation rate still varies seasonally, which means that $\sigma$ also varies in time for any firn layer (again, even in steady state). Finally, the reader has no information about how $r_e$ is calculated in the computations of these steady state strain rates.
Furthermore, the "*regularized least square algorithm*" is not described. Why not proceed to a simple least square? What is the penalty term? What are the penalised factors? And, most importantly, which coefficients are subject to the optimisation and what is the range of possible values covered by the optimisation?

I think that the authors can easily understand that the issues I raise here are concerning with respect to the GMD review criteria.

2) The ELM firn density simulations

My first concern relates to the data that is used for model evaluation. Why use the cores of Mosley-Thompson et al. (2001) and Lamorey (2003)? And why do the authors average the Greenland cores? They highlight themselves that "*variability can be large, particularly across the GrIS*" (line 337). Why not compare an observed firn core to the model simulations for the grid cell of the corresponding location? Averaging observed and modelled firn depth-density profiles makes little sense. The authors themselves seem to point out this shortcoming of the study (lines 318-320): "*though our analysis with ELM thus far is limited to a generalized comparison with a broad (climate) perspective rather than to a more site-specific comparison against direct observations*". So why was a site-specific comparison not performed?

Moreover, the data selected is not in line with the objective of the study: improving dry densification schemes. Most of the Greenland grid cells are likely affected by melt, and Siple Dome is an area of Antarctica with relatively high melt rates for the continent. Why don't the authors select data only from dry snow areas (higher accumulation zone of Greenland and more inland regions of Antarctica)? Do the authors know about the extensive SUMup dataset (Koenig and Montgomery, 2019) that includes many more firn cores? The occurrence of melt is clear because there is "*formation of ice lenses*" (line 241). But no information is provided about the model schemes for simulation of meltwater percolation and refreezing. Moreover, the simulations are performed with atmospheric forcing at very coarse resolution, which is underlined at lines 342-347 (I mention here that the resolution is not provided in the manuscript). This forces the authors to artificially adjust their evaluation: "*this large grid cell remapping lead to a cold bias, resulting in too-slow densification. Therefore, we adjusted our Siple Dome comparisons to include grid-cells away from the coast that better represent atmospheric conditions and result in a more realistic density simulation*". Firstly, I would think that grid cells away from the coast should be even colder and thus enhance the cold bias. Secondly, this further underlines the question of why choosing Siple Dome and Greenland firn cores to evaluate the models. This choice brings in problems related to the adequacy of the model forcing, which makes it very difficult to disentangle firn model deficiencies from errors due to inadequate forcing.

The spin-up period is taken to be 260 years. This is likely too short for low-accumulation grid cells (thus most of the dry snow zone) to build a full firn layer (i.e. until ice density is reached). The authors should thus support their statement (lines 226-227) that the profiles "*averaged from the final 100 years of simulation results*" (thus starting the averaging only after 160 years) are "*steady-state density profiles*". For example, after 160 years of an accumulation rate of 0.07 m w.e. yr$^{-1}$ (the limit assumed for warm dry snow zones in Section 3.2), only 11 m w.e. have accumulated, which corresponds roughly to a firn column of 20 m. I doubt that this represents a steady state. As shown in Figure 4 (T=-34°C), firn that is 100 years old is only at 600 kg m$^{-3}$ density, showing that the firn layer is most likely not in steady state after 160 years and not even after 260 years. Concerning the fresh snow density, the A'76 model calculates surface densities by itself, while vK'17 and vK'17+ use a fresh snow parameterisation (not given in the manuscript…). But how is fresh snow density calculated for the A'10 model? The prognostic equation for $r_e$ should also be given or referred to.

The approximation of vertical strain rates (line 244) is also unclear. This raises the same questions that I mentioned above about the dynamic variables when assuming a steady state. In my view, the authors should include a detailed explanation about how the steady state strain rates of A'76, A'10, vK'17 and vK'17+ are calculated. This holds for both the calibration step as for the ELM simulations (i.e. the values appearing in Figure 4).

When analysing and evaluating the results, there is a severe lack of quantification. This holds for both the Equilibrium climate simulations and the Twentieth century climate simulations. I give some examples:

- "*the semi-empirical model improves the density profile*" (line 260): improves with respect to what (I guess that the authors mean A'10 improves the density profile with respect to A'76)? And the statement of "improvement" should not be based on a mere visual comparison of Figures 2 and 3. Moreover, it should be clarified that the authors evaluate the model results against results from HL, which is not a guarantee of model accuracy.

- "*Densification tapers-off at lower densities (around 450 kg m$^{-3}$) for colder climates, a temperature-dependent effect enhanced with the model from Arthern et al. (2010).*" (lines 248-250): from Figures 3 and 4, it is not obvious that this effect is tronger in A'10 than in vK'17+. The enhancement of the effect should thus be quantified.

- "*A lower model variance occurs when it does not covary with the empirical model. This effectively reduces a model's prediction risk if it does not also result in an increased bias.*" (lines 303-304): If the authors discuss about the bias of models, they can simply add a "Bias" column in Table 2.

- "*both models show improvement compared to their original counterparts (ELM v1 and CLM).*" (lines 337-338): this is impossible to evaluate for the reader because (1) only the results of A'10 and vK'17+ are shown in Figure 6, and not

the ones of their so-called "*original counterparts*" (which are A'76 and vK'17 I suppose), and (2) there is no quantitative evaluation of the models' performance with respect to the observed data (e.g. RMSE, bias, etc.).
- "*Encouragingly, our simulation results compare well with firn density measurements and indicate an improved capability in the ELM.*" (lines 348-349): same remarks as for the previous point.

3) The novelty and objective of the study
GMD review criterion:
    *Does the paper present novel concepts, ideas, tools, or data?*
Firn model optimisation has been addressed in numerous studies over the recent years (e.g. Ligtenberg et al., 2011; Kuipers Munneke et al., 2015; van Kampenhout et al., 2017; Smith et al., 2020; Verjans et al., 2020). Four of the studies mentioned have already investigated the optimisation of the model of Arthern et al. (2010), for Antarctica, Greenland or both. An easy and straightforward way to improve the ELM would be to implement the parameterisations developed in these studies. If the authors want to address the same problem, they should propose a new, original method. However, in contrast to the existing literature, they do not calibrate the model of Arthern et al. (2010) with observations but with HL-computed strain rates. And, as mentioned above, it is unclear to me how they calibrate a dynamic model to steady state strain rates. They should justify why their methodology is better suited to their objective than using what other researchers have already accomplished. Moreover, the objective stated in the conclusion of improving the capability of the ELM to better simulate refreezing rates in firn is not in line with the study itself. The focus of the calibration is on dry firn densification and does not support the statements at lines (377-381): "*With an evaluation of the simulation of dry firn densification, we have optimized the ELM firn model for future studies of the impacts of liquid water on firn density and SMB. Ultimately, this study seeks to enable better predictions of SLR as a direct result of surface melt and mass loss from the GrIS.*"
If the authors do want to better capture liquid water effects, they should focus on this very challenging topic by studying the mechanisms of wet firn compaction, meltwater percolation and refreezing.

4) The clarity of the manuscript
GMD review criterion:
    *Is the overall presentation well structured and clear?*
It is very difficult for the reader to understand the different steps of the study. The authors alternate between different ways to refer to a same thing. For example, the A'10 model is sometimes referred to as "*A'10*" and sometimes as "*the semi-empirical model*", the vK'17 model is sometimes referred to as "*vK'17*", sometimes as "*the CLM*" and sometimes as the "*Snowpack model*" (see Figure 5). Similarly, it is never clearly stated that the "*empirical strain rates*" correspond to the ones computed with the HL. A first, simple way to improve the clarity would be to consistently use the terms A'76, A'10, vK'17 and vK'17+ throughout the manuscript, including in the captions of the Figures. In line with this, the Section 3.1 should be split in four subsections that clearly detail each of these four models instead of subsections presenting equations which are subsequently assigned to the models in a confusing way.
The Figures and Tables also lack clarity. In Figures 2 and 3, why are high values of accumulation only shown at depths greater than 15 m? Even in a high-accumulation climate, there will always be a shallow and a deep part of the firn column. And how were the surface density values chosen for the HL-computed profiles? The caption of Figure 1 mentions that the new firn model "*can extend as deep as 80 m*", whereas it is always presented as a "*semi-infinite*" grid in the text. Which of these two statements is true? In Figure 4, why are there points without a vertical error bar? And why do some points have a horizontal error bar (age should be well-determined for any firn layer of any model run)? In Table 1, equation numbers could be provided for each model to know which equations apply to which model. In Table 2, the model names should be used in the column "*Densification model*" instead of the mechanisms applied and the variable for which the statistics are calculated should be specified in the caption (presumably strain rate values). Improving the structure of the manuscript could possibly help decrease the degree of confusion for the reader when trying to understand the study.

Specific comments:
line 2:
Change "*consist*" to *consists*.
line 15:
I doubt that any paleoclimate study uses Earth System Models to determine pore close off depth and timing.
line 25:
Repetition of "*coupled*".
lines 32-35:

As far as I understand, there is a contradiction between "*does not yet exist*" and explaining the implementation of the advanced firn model in the CLM.

lines 47-48:

Change "*those predicted by Herron and Langway (1980)*" to *those predicted by the model of Herron and Langway (1980)*. Moreover, I suggest using a consistent way to refer to this model (e.g. HL'80).

lines 52-55:

Add an explanatory sentence about the fact that snowpack models and firn models also have a different vertical scale of application.

Section 2.1:

Provide units of all the variables and quantities presented. This will make clear that there are some unit inconsistencies in some of the equations (which I give below).

Equations 1, 2, 3 and 4:

The variables $\dot{\varepsilon}$ and $\left(\frac{1}{\Delta z}\frac{\partial \Delta z}{\partial z}\right)$ are equivalent to each other as far as I understand. Use either one of the two notations.

line 78:

Specify if $\left(\frac{1}{\Delta z}\frac{\partial \Delta z}{\partial z}\right)_{dm}$ is also considered in CLM (v5).

line 82:

In CLM(v5), $c_\eta$ = 358 kg m$^{-3}$ and $f_2$ was set to 4. Only $f_1$ accounts for the effects of liquid water and not $c_\eta/(f_1 f_2)$.

line 89:

The characteristic depth is not "*a single valued proxy for a given site's full density profile*" but only for the upper density profile.

line 89:

No s at *stages*.

line 89:

Add here the explanatory sentence about why models assume a two-stage densification process.

line 92:

The sentence "*Empirical firn densification models typically employ analytic functions that assume a steady-state density profile*" is not true. Only the model of Herron and Langway (1980) and a few others provide analytic functions but almost all of the recent firn models are dynamic models.

line 93:

Rephrase.

Equation 5:

This equation is erroneous. The units of the left- and right-hand sides do not match. The correction is: $w(z) = \frac{A}{\rho(z)/\rho_w}$, where $\rho_w$ is the density of water (1000 kg m$^{-3}$).

Equation 7:

Again, this equation is erroneous and there is a unit inconsistency. The correction suggested above, fixes the error.

Equation 8:

The variable $P$ is defined here as the "*overburden pressure*", which makes it equivalent to $\sigma$. I suspect that this variable corresponds to the $P$ as defined in Equation 9, which should be called the *grain-load stress*. I underline here that in the model of Arthern et al. (2010), it is really $\sigma$ that is used and not the grain-load stress. The authors should explain why they differ from the original model of Arthern et al. (2010) on this point.

lines 176-177:

Specify that the "*plausible firn density-versus-depth profiles*" were computed with the different models and the HL.

Section 3.2:

Why do the authors decide to draw annual temperatures from a distribution representative of the global Earth climates instead of the polar climates? Is the objective to have much more values close to T=-25°C? This should be clarified. How are all these values decided:

- -25°C as a threshold (whereas ELM simulations involve warmer sites)
- -51°C as limit between low- and high-accumulation sites (many sites can have T>-51°C and A<0.07 m SWE yr$^{-1}$)
- surface density values between 300 and 380 kg m$^{-3}$

line 196/

Specify the resolution of "*coarse-resolution*".

line 197:

What does "*(an "I-compset" at ne11 resolution)*" mean?

line 199:

Change *January 1st* to *January 1st 1901*.

line 206:

Define "*restart runs*". In general, it is good practice to define any term used that may not be straightforward to everyone reading the study.

Table 1:

Add a column for $\rho_{dm}$ values. Add another column that indicates which equations apply to which model, with the corresponding equation numbers (see Major Comment 4).

lines 209-215:

All the details provided about the observational data should be given in a separate subsection. Are the sites of measurements affected by surface melt? And are these firn core measurements open access?

Section 4:

There are a lot of speculative statements in this section. I suggest splitting it into a section Results and a section Discussion, so that the reader can distinguish between model results and the thoughts of the authors.

line 219:

Change *accumulations* to *accumulation rates*.

lines 217-222:

Here, the entire methodology is again defined. This can be confusing for the readers. For example, I suggest rephrasing the sentence "*To improve the accuracy of our firn model simulations, we optimize compaction terms against empirical strain rates using statistical modeling*" because this was the point of a previous section.

line 229:

Does the statement "*the mean annual temperature is within a couple degrees of -25 °C*" refer to the results of Figure 2 for T=-27°C? If so, it would be clearer to give the exact mean annual temperature value.

lines 236-237:

"*These simulations demonstrate a stronger effect of temperature on densification rates, resulting in more variation in density with depth (Fig 3).*" I think that this sentence means more variable depth-density profiles according to the mean annual temperature. If so, it should be rephrased.

lines 242-243:

Specify which models use the "*better fresh snow density parameterization*".

lines 260-261:

Strange use of commas.

line 264:

What is "*over-densification*"?

line 265:

The notion of "*density profiles that vary too weakly with depth*" should be replaced with the one of density that increases too weakly in depth.

line 265:

What does "*Their*" refer to?

lines 274-277:

Are these the only coefficients included in the optimisation? Or the only ones that were decided to be changed? See Major Comment 1.

line 276:

The coefficient $f_2$ is related to the grain radius (see Vionnet et al., 2012). If the ELM calculates grain size, $f_2$ can be calculated accordingly. It is crucial that the authors clarify why they decide not to follow the original formulation of $f_2$ (as a function of grain size) but to consider it as a pure tuning factor. This is all the more relevant since it is emphasised throughout the manuscript that firn models need to account for microphysical features.

line 286:

Specify that the density model coefficients are calibrated to the HL-computed strain rates.

lines 288-290:

Specify the variable for which the statistics are calculated. See Major Comment 4.

lines 293-298:

How do the authors explain these results?

Table 2:

The mention to eq. (5) is an error because it is not the one for destructive metamorphism and $c_5$ does not appear in this equation.

line 300:

The value of $R^2$=0.67 is valid only for strain rates in the second stage. Note also that Table 2 shows $R^2$=0.66.

line 303:

"*A lower model variance occurs when it does not covary with the empirical model*". This statement sounds like a general statement, but I believe that it is applicable only to the results of this specific study.

line 306:

What does "*these results*" refer to? The paragraph above is about variances in the compaction rates.

lines 306-307:

"*negative correlations between overburden pressure and empirical strain rates*": this is explained by higher overburden being applied to deeper firn, which is at higher density and thus compacts less. A simple explanation could be provided to the reader.

lines 311-312:

Note that Equations 3, 4 and 8 are directly dependent on density.

lines 320-328:

Are these results or speculations? Where do these values come from? And did the "*statistical computing*" focus only on these specific coefficients of A'10? Such conclusions should not be stated without quantitative results to support them. I emphasise again the need to clarify the optimisation method and its results.

lines 331-332:

Why did the authors decide to use the optimised vK'17 but not the optimised model of A'10?

line 333:

Typo "*the the*"

line 341:

"*ne11*" is not defined.

line 350:

The statement "*we should focus on the near surface layers, as they contain the primary SMB components*" should be clarified.

line 351:

"*it could be necessary to model the upper most 20 m*": Is this figure of 20 m supported by the results? If not, references should be provided.

lines 352-353:

"*the optimized version (vK'17+) is likely the better choice for implementation into the next major release of the E3SM*": Again, it is unclear if this is speculative or if the results provide strong evidence for this. And how would an optimised version of A'10 compare to vK'17+?

lines 353-355:

A low bias is usually preferable to a high bias. What is meant by this sentence?

line 364:

"*In the near future, this could be the entire GrIS*": What is the "*near future*"? And references should be provided.

Conclusions:

See Major Comment 3.

line 370:

Change "*with steady-state empirical models*" to *with a steady-state empirical model*.

lines 370-371:

I do not believe that the analysis is "*similar to that by van Kampenhout et al. (2017) for CLM*".

Code and data availability:

The availability of the firn core measurements should be provided in this section.

Figures:

See Major Comment 4. Also, in all Figure captions and legends, the models should be consistently referred to as A'76, A'10, vK'17 and vK'17+.

Figure 1:

The vertical extension of "*80 m deep*" (caption) contradicts the "*semi-infinite*" stated in the text.

Figures 2 and 3:

The vertical/horizontal lines at $\rho$=550 kg m$^{-3}$ and $\rho$=550 kg m$^{-3}$ can be removed to improve the clarity of the Figures. Why do high accumulation rate values appear only at great depth?

Figure 4:

"*for various plausible accumulation rates (0.11–0.50 m SWE yr.$^{-1}$)*": Why is 0.11 m SWE yr.$^{-1}$ chosen as lower limit? The construction of the horizontal error bars is not clear to me.

Figure 5:

Do these plots show the results of A'10 (top) and vK'17 (bottom)?

---

## Referee Comment (RC2) · Anonymous Referee #2 · 11 Sep 2020

This paper addresses a specific, pressing need to improve the simulation of firn in dry-snow zones of ice sheets in the E3SM. The authors present a compelling study of firn-density model improvements, ultimately achieving the first step toward implementing improved firn simulations in the ELM. Their work highlights the need for two models of firn densification, above and below the characteristic-density depth, to achieve simulated density-depth profiles as observations made in the field, and suggests the required next steps for using these models to generate better predictions of sea level rise. Below is a list of general and specific comments for the authors to consider in revising this manuscript.

[Figure]

General Comments: - This is a very well written manuscript that is motivated and articulated clearly. I am not an expert in ESMs or their various component models, and therefore cannot comment on the authors' specific implementation of firn compaction models into the E3SM Land Model.

- The stated goal of this work is to more accurately simulate snowpack evolution in the ELM, including over the Greenland and Antarctic ice sheets. With this goal, why do the authors increase the snowpack from 1m (in the previous version) to up to 60 m in this version? In the dry snow zone, it's common for firn column to vary in depth from 50 to 120m (Cuffey & Paterson, 2010), depending on site conditions. Many sites have firn columns deeper than 60 m (and even the 80 m depth that Figure 1 indicates is possible), and therefore this estimate of snowpack evolution won't be valid for large swaths of the Greenland and Antarctic ice sheets. If instead the authors are referring to depths in m SWE, this should be made explicit within the text and in the figure axes.

- In various places the authors describe "improvements" or "slight improvements" between models, especially in Seciton 4.1. How do the authors determine these improvements? Are they quantifiable?

Specific Comments:

Section 3.1 – The abstract states that the authors improve the depth of snowpack in the ELM from 1 m to up to 60 m, while the Figure 1 caption states that the new model layers can extend to 80 m. Why is there a discrepancy between these two extended depths?

Section 3.3.2 – a brief description of the Greenland sites used from the Mosley-Thompson et al. (2001) study would be helpful here. There were quite a few cores in that thorough study. How did the authors decide which cores to average into a composite GrIS density profile? The accumulation rate varies quite a bit across Greenland, especially north to south. Were northern and southern sites averaged together?

Additionally, only sites near Siple Dome were used in Antarctica. Since sites in East Antarctica has much lower mean annual temperatures and accumulation rates that Siple Dome, it may be more appropriate to claim that this applies to West Antarctica than both Antarctic ice sheets.

Section 4.1 – Lines 237-239: It would be helpful to the reader if the authors indicate which model they're referring here when they say "…this dynamic implementation of eq. (8)…" and "…the original compaction parameterization, from eq. (3)." Are these referring to models vK'17+ and A'76, respectively?

Lines 225-227: What is the justification for using the 1901-1920 reanalysis data to generate the steady-state density profiles? Is this considered an average period of time? If so, what metric was used to determine that it was best to use the data from 1901-1920?

Lines 225-227: The spin-up of the model simulated 260 years of snow accumulation is adequate for creating a typical dry snow-zone firn column in Greenland but would not reach back far enough to erase the natural firn density profiles in East Antarctica. Why was a spin-up of 260 years chosen?

Lines 227-229: are the authors referring to two of the examples given (-27C and -20C) here? Describing results for scenarios with mean annual temperature "within a couple degrees of -25C" is imprecise and leaves the reader wondering if they're missing a panel in Figure 2 for the -25C scenario (and other scenarios in between).

Lines 229-230: It'd be helpful to indicate here that the results from the Herron & Langway model are the colored bars in Figure 2.

Lines 237-239: In the discussion of Figure 3 here, the authors describe that the dynamic implementation of eq. (8) in the ELM (Again, is this the vK'17?) results in characteristics depth (550 kg/m3) more consistent with Herron & Langway for T > -32C, but the authors do no present any of the results for the T=-32C scenario. How was this

cutoff determined? Additionally, the panel for the T = -20C scenario shows that neither A'10 nor vK'17+ do a good job predicting the characteristic depth. Therefore, how do the authors conclude that the results are more consistent with H&L for T > -32C?

Figures 2 & 4: which lines are the authors referring to in these captions when they say "Line graphs show 100-year means from ELM simulations. . ."? All of the lines (except dashed & dotted?)? This vague statement is confusing due to the number of lines plotted.

Figure 2 – how are the range of surface densities used in the empirical modeling shown in the figure?

Figure 3 – comparing the steady-state density profiles generated for the 3 mean annual temperatures to sites with similar conditions (Table 2.2, Cuffey & Paterson, 2010), it appears that the ELM and empirical modeling results reach pore close-off density at too shallow of a depth for -20C, empirical modeling results.

Figure 4 – The stated goal for this figure is to better understand what drives rapid densification near the surface. Therefore, it would help the reader to have a second x-axis displaying the depth. Since we don't have a depth-age scale, it's hard to interpret where the near-surface is in this figure. What causes the jump in vertical strain rate in the empirical modeling results (colored lines)? Additionally, the error bars make it very hard to see the black and grey line trends. Consider altering the error bars in some way to allow the lines to become more visible. In the lower panel, why do the ELM simulation lines begin near 40 years instead of 0?

Technical Corrections: L10: remove 'when' from '. . .compared to when using. . .' L25: 'coupled' twice in this sentence, sounds awkward L34: need 'it' after 'implemented' here L225: the authors start what "with a baseline configuration. . ."? L260: don't need comma after "ELM" Figure 1: describe variable 'z' in the caption Figures 2 & 3: the bottom row of figures overlaps the y-axis units, space these out slightly so that each axis is legible. Figure 4: the top row of figures overlaps the y-axis units, space these

out slightly so that the axis is legible. Figure 6: add description of density measurements are shown in blue, as well as that the dotted and dashed lines represent the characteristic density and pore close-off density, respectively, to the caption.

---

## Referee Comment (RC3) · Anonymous Referee #3 · 16 Sep 2020

The manuscript by Schneider et al. is concerned with improving the simulation of snow and firn in the E3SM Land Model (ELM), in particular snow density. The subject is timely and fits the purpose of GMD well. The topic is also relevant, as firn acts as a major control on the surface hydrology and surface mass balance of ice sheets, which is relevant when coupling dynamical ice sheet models with Earth System Models. For this reason, I was happy to learn that E3SM is developing into this general direction. The authors demonstrate good knowledge of the literature, and they tested and recalibrated models for their purpose, which I applaud. The quality of the figures is good and the writing as well. Unfortunately, I have three major concerns with the study, which I will lay out below.

[Figure]

The first concern is about the statistical modelling approach explained in Sect 3.2. I didn't get very warm feelings about this. For instance, in the direct comparisons carried out in Sect. 4.1.1 and Figures 2-4, output from a coarse resolution ELM simulation with 6-hourly CRU-NCEP forcing is compared to steady-state profiles from the Herron & Langway model with idealized (synthetic) forcing. To me this feels like comparing apples to pears. Unnecessarily, it seems, since the ELM (like CLM) can also be forced with synthetic data in single-column mode, offering a more direct comparison. After all, to quote Arthern et al. (2010), "Changes in weather and climate can cause temperature, accumulation rate, and depositional density to vary. Consequently, and in violation of Sorge's Law, the density profile r(z, t) will fluctuate with time t." It is to be expected that this variation causes differences mainly in the upper ∼10 m of the firn pack, i.e. the active layer, which is unsurprisingly where the largest differences between the Herron&Langway model and the ELM simulations is found (Figure 2 & 3). Arthern2010 further notes that "Alternative models, broadly based upon the Herron and Langway [1980] parameterization, have employed different formulations for the sensitivity to temperature [Li and Zwally, 2004; Helsen et al., 2008]". This quote is a hint that the temperature dependence of Herron & Langway is not to be taken as the truth, which is kind of what the authors seem to be doing in Section 4.1.2, but also in the next section (4.1.2) where they calibrate model coefficients using their synthetic HL density profiles. I feel the heavy reliance on the HL model and synthetic data isn't properly justified.

My second concern is with the readability of the manuscript, and in particular the range of different model configurations that are presented, and the purpose for all of them. The title of the paper is "Snowpack and firn densification in the E3SM", however at the end of the manuscript I'm lost to which results are now representative of the improved E3SM and which aren't. Line 184 seems to suggest that the coefficients in the new E3SM are optimised from one of the other models, however it isn't stated which one, and the final configuration in E3SM is not named. The wording in Line 320 ("might expect") and Line 376 ("a first step") is also contradictory to this, suggesting that the

coefficient optimization isn't really applied at all in E3SM. Does that mean that the entire Section 4.1.2 is actually superfluous and could be removed? Are none of the models discussed in this paper actually adopted in E3SM? I encourage the authors to make this more explicit. Alternatively, the authors could choose to take the focus off E3SM and shape it into a more general firn-modelling paper, I'll leave that up to them.

My third major concern is with the comparison to observed firn core data (Section 4.2). Here the authors aggregate ELM data from across the GrIS from a coarse resolution simulation, and compare this a single point measurement (or at least, an approximation to this). Again, this seems to me like comparing apples to pears and a pretty crude approach. For instance, there will most certainly be grid cells in the composite that experience melt, whereas the interest is on dry firn compaction. Can the ELM not be forced in single-column mode with high-resolution meteorological data from e.g. ERA5, or a high resolution run with E3SM, more approximate to the actual weather at the site?

All in all, I find this paper not convincing in its methodology, and I feel further justification or experiments are needed.

Specific comments

L61: pressure not pressures

L130: Actually, Muntjewerf et al. (2020) provides little detail on their model setup, and none on snow/firn modelling. Suggest to remove, and optionally replace with the following reference, which does actually provide more detail on snow modelling within CESM:

van Kampenhout, L., Lenaerts, J. T. M., Lipscomb, W. H., Lhermitte, S., Noël, B., Vizcaíno, M., et al. (2020). Present-Day Greenland Ice Sheet Climate and Surface Mass Balance in CESM2. Journal of Geophysical Research: Earth Surface, 125(2), e2019JF005318. https://doi.org/10.1029/2019JF005318

L143: This title suggests that the new surface density scheme is only applied over ice

sheets. However, this is not explained anywhere, so the title should be changed?

L180: Could you comment on what basis the values for T and A were selected? In Line 220 you define the "dry snow zone" as 0.5 m SWE / year, whereas here the value of 0.4 appears.

L195: Could you comment on the quality of the meteorological data in CRU-NCEP over the regions of interest, i.e. ice sheets? Do you think the outcome of the simulations depends a great deal on the choice of meteorological forcing ?

L196: please specify what nominal resolution (in degrees or km) does the ne11 resolution correspond to?

L215: Since this manuscript concerns a global model, and there are only 16 layers to begin with, two reference firn density profiles are probably justified. Just be aware that there are more firn cores out there, e.g. see Figure 1 in the recently published TC paper by Verjans et al. :

Verjans, V., Leeson, A. A., Nemeth, C., Stevens, C. M., Kuipers Munneke, P., Noël, B., & van Wessem, J. M. (2020). Bayesian calibration of firn densification models. The Cryosphere, 14(9), 3017–3032. https://doi.org/10.5194/tc-14-3017-2020

L260: remove comma after ELM ?

L 373: near-surface firn densities that are too low, not large?

L 376: E3SM Project: not clear, is this a reference?

L 380: and AIS? Surface melt is believed (or known) to be important for the stability of ice shelves.

Reference list: to avoid cluttering, I'd suggest to remove the URLs and replace with DOI where needed.

L 446 : fix title L 483 : fix title L 512 : fix title

**[GMDD](https://doi.org/10.5194/gmd-2020-247)**

Interactive
comment

Figure 4: The caption describes the meaning of the line graphs in the first row, but not the crosses used in the second row.

Figure 5: The titles of the subfigures could be made more informative. Also, the colour bar could be made more restrictive, it appears densities < 300 kg/m3 do not occur at all.

Figure 6: The legend appears not consistent with the previous figures, e.g. ELM-A'10 instead of just A'10.
* * *

---

## Author Comment (AC1) · 13 Nov 2020

13 Nov 2020

re: "Review of Schneider et al. 2020", Vincent Verjans, 27 Aug 2020

We appreciate your thorough review and are working to address your concerns. Following below is our response (in blue) to your enumerated major and specific comments, which are italicized here for reference:

1. *The lack of information about the optimisation*
   *Here are two review criteria of GMD:*

   - *"Are the methods and assumptions valid and clearly outlined?"*

   - *"Is the description sufficiently complete and precise to allow their reproduction by fellow scientists (traceability of results)?"*

   *I focus first on the optimisation method applied (Section 3.2). The entire optimisation method is described in a single sentence (lines 184-186):*
   *"From our estimated empirical strain rate-versus-depth data, we optimized the previously described densification model coefficients (from A'76, vK'17, and A'10) by applying a regularized least squares algorithm for two stages of densification (above and below $\rho$ = 550 kg m$^{-3}$)."*
   *The A'76 model includes 7 different coefficients ($c_1, c_2, c_3, c_4, c_5, \rho_{dm}, \eta_0$), the vK'17 includes 10 ($c_3, c_4, c_5, \rho_{dm}, \eta_0, c_\eta, f_1, f_2, a_\eta, b_\eta$) and the A'10 includes 7 ($c_3, c_4, c_5, \rho_{dm}, k_c^{\rho<550}, k_c^{\rho>550}, E_c$). Additionally, in the Results section, the authors mention "adding a constant compaction term" (line 277), which does not appear in any equation and is not explained in the Methods section. Throughout the manuscript, the authors never state which coefficients are subject to the optimisation. Moreover, they mention in Section 4.2 that "we have yet to test in ELM an optimized version of the semi-empirical model". From my understanding, the "semi-empirical model" is A'10 and they decided to compute the ELM simulations with the original version of A'10 and not the optimised version. In contrast, for some reason, the authors did the ELM simulations with the optimised version (vK'17+) of vK'17. They still provide speculative avenues for a re-parameterisation of A'10 at lines 322-328. These statements are not supported by any quantitative information about a better fit of the optimised A'10 either to observed data or to the strain rates generated from the model of Herron and Langway (1980) (referred to as HL hereafter). Finally, they assert that they optimise A'76 ("we optimized the previously described densification model coefficients (from A'76, vK'17, and A'10)"). However, the only information to be found about the modifications brought to the model is the change of the value of $c_5$, but nothing about other parameters and nothing about a performance comparison between the original A'76 and the optimised version.*
   Clearly, we failed to provide essential details regarding the optimization method used to calibrate firn densification model coefficients. While we experimented with numerous model configurations, the optimization referred to in the manuscript pertains to $c_3$, $\eta_0$, and $k_c$, as referred to above, plus an additional term not mentioned in the manuscript! We appreciate you pointing out this oversight. These coefficients were calibrated using a convoluted least squares algorithm that we must clarify in our revision.

At the time of submission, we did not have ELM simulation results from the optimized version of the A'10 configuration (hereafter A'10+). Ideally, we would test dozens of model configurations in ELM. However the ELM simulations on century timescales are computationally expensive and time consuming. Based on the preliminary results of our statistical modeling, we experimented with a small selection of model configurations (including both vK'17 and vK'17+), which we describe in the manuscript. Our speculative reparameterization of the A'10 model has now been tested in a century-scale ELM simulation, but due to the noted lack of clarity, it is not clear how we arrived at these speculations. Finally, we discovered that the original (A'76) configuration implemented in ELM could be significantly improved by simply modifying $c_5$. Therefore, we calculated the optimal value of $c_5$ using our statistical model, but again, these results are impossible for the reader to fully comprehend due to our lack of clarity.

To address these critical flaws, we are expanding Section 3.2 by adding details regarding the optimization method. We are also adding new results from the A'10+ ELM simulations and are removing the speculative comments toward the end of Section 4.1.

*Coming back to the optimisation methodology itself, the authors decide to select annual mean temperatures only below -25°C, but they then proceed to model simulations for grid cells where the annual mean temperature is as high as -20 °C (Section 3.3, Figures 2, 3 and 4). It is puzzling that the authors themselves suggest a better approach to selecting mean annual temperatures and accumulation rates, which would be easy to implement (lines 357-361). They also claim to calibrate the models by matching the computed strain rates to the HL strain rates. The issue is that several models use dynamic variables in the strain rate equations: $T, \sigma, P$ and $r_e$ in Eqs 1, 2, 3, 4, 8, 9 and 10. The HL can provide analytic solutions of strain rates for steady state annual mean temperature and accumulation. The dynamic models require values for the dynamic variables at each time step and for each layer of the firn column. A steady state annual mean temperature does not correspond to all firn layers having the same temperature year-round (temperature still varies seasonally when in steady state). Similarly, accumulation rate still varies seasonally, which means that $\sigma$ also varies in time for any firn layer (again, even in steady state). Finally, the reader has no information about how $r_e$ is calculated in the computations of these steady state strain rates.*

The calibration method only considers density profiles for mean annual temperatures less than -25°C so that it will be optimized for dry firn densification. However, ELM simulations provide global results, including warmer grid-cells that are shown in Figs. 2, 3, and 4. In the future, it would be interesting to download output from regional climate models and reanalyses to read into the statistical model. However, such an approach is beyond the scope of this study. Regretfully, we failed to provide enough details about the rest of our statistical model and optimization method including how we calculate the dynamic variables $T, \sigma, P$ and $r_e$.

To address these issues, first, we will change the comparisons of ELM results versus Herron and Langway (1980) so that the columns in Figs. 2, 3, and 4 center around mean annual temperatures of -39, -32, and -25 °C. This will eliminate the discrepancy in the temperature cutoff referred to above. And second, we are expanding Section 3.2 to include details regarding how dynamic variables ($T, \sigma, P$ and $r_e$) are estimated.

*Furthermore, the "regularized least square algorithm" is not described. Why not proceed to a simple least square? What is the penalty term? What are the penalised factors? And,*

*most importantly, which coefficients are subject to the optimisation and what is the range of possible values covered by the optimisation?*

Yes, the least squares algorithm is convoluted and totally obscure based on the manuscript alone. None of these reasonable questions are answered and therefore the results cannot be reproduced.

To correct this critical flaw, we are adding to Section 3.2 or an Appendix a more detailed description of the algorithm including all pertinent tuning parameters and coefficients subject to the optimization.

*I think that the authors can easily understand that the issues I raise here are concerning with respect to the GMD review criteria.*

Yes, the issues you raise show that the originally submitted manuscript lacks the clear explanation needed to make it reproducible. The forthcoming revision will resolve these issues as we describe.

2. *The ELM firn density simulations*

   *My first concern relates to the data that is used for model evaluation. Why use the cores of Mosley-Thompson et al. (2001) and Lamorey (2003)? And why do the authors average the Greenland cores? They highlight themselves that "variability can be large, particularly across the GrIS" (line 337). Why not compare an observed firn core to the model simulations for the grid cell of the corresponding location? Averaging observed and modelled firn depth-density profiles makes little sense. The authors themselves seem to point out this shortcoming of the study (lines 318-320): "though our analysis with ELM thus far is limited to a generalized comparison with a broad (climate) perspective rather than to a more site-specific comparison against direct observations". So why was a site-specific comparison not performed?*

   The evaluation against observations was also challenged by the other reviewers.

   To redeem this flaw, we are adding to Section 4 a new analysis that evaluates Greenland Ice Sheet (GrIS) results against the SUMup dataset (Koenig and Montgomery, 2019), as you suggest below. This analysis covers the accumulation areas of the GrIS better than the subset of measurements provided by Mosley-Thompson et al. (2001) and adds a geographical comparison with a quantitative evaluation. By computing a similarity matrix, we categorize measurements nearest in distance and time to corresponding ELM simulations and then assess model accuracy with root mean squared errors (RMSE). This new analysis serves two objectives: one, improve the quality of the model to observational comparison by controlling for space (location) and time (year); and two, offer metrics for quantitatively assessing model accuracy.

   *Moreover, the data selected are not in line with the objective of the study: improving dry densification schemes. Most of the Greenland grid cells are likely affected by melt, and Siple Dome is an area of Antarctica with relatively high melt rates for the continent. Why don't the authors select data only from dry snow areas (higher accumulation zone of Greenland and more inland regions of Antarctica)? Do the authors know about the extensive SUMup dataset (Koenig and Montgomery, 2019) that includes many more firn cores? The occurrence of melt is clear because there is "formation of ice lenses" (line 241). But no information is provided about the model schemes for simulation of meltwater percolation and refreezing. Moreover, the simulations are performed with atmospheric forcing at very coarse resolution,*

*which is underlined at lines 342-347 (I mention here that the resolution is not provided in the manuscript). This forces the authors to artificially adjust their evaluation: "this large grid cell remapping lead to a cold bias, resulting in too-slow densification. Therefore, we adjusted our Siple Dome comparisons to include gridcells away from the coast that better represent atmospheric conditions and result in a more realistic density simulation". Firstly, I would think that grid cells away from the coast should be even colder and thus enhance the cold bias. Secondly, this further underlines the question of why choosing Siple Dome and Greenland firn cores to evaluate the models. This choice brings in problems related to the adequacy of the model forcing, which makes it very difficult to disentangle firn model deficiencies from errors due to inadequate forcing.*

To restrict our evaluation to grid-cells without melt, we originally removed ELM columns that experience mean annual temperatures of greater than -25°C. Regarding the Siple Dome case study, we were mistaken about the "...cold bias, resulting in too-slow densification" (line 345). Rather, ELM has a "warm" bias there, resulting in too much densification, so we adjusted the ELM comparison inland, which is more representative of the actual Siple Dome climate and resulted in better agreement with the observed density profiles. Your point about the difficulty in evaluating course resolution ELM grid cells to point based measurements is why we generally fall back to the model of Herron and Langway (1980) to evaluate model performance. The HL empirical model can be applied to gridcells that experience faulty atmospheric input data. The comparisons plotted in Figs. 2, 3, and 4 control for mean annual temperature and accumulation to help disentangle firn model deficiencies from inaccurate atmospheric forcing. As mentioned above, we now have a much more comprehensive evaluation against the SUMup (measurement) dataset (Koenig and Montgomery, 2019).

Because both original evaluation methods are questionable, we are adding to Section 4 another evaluation that better controls for geographic location and time and also provides RMSEs with reference to SUMup observations (Koenig and Montgomery, 2019). This new evaluation indicates that our optimization method improves upon the snowpack and firn compaction model currently in CLM for most model grid-cells across the GrIS.

*The spin-up period is taken to be 260 years. This is likely too short for low-accumulation grid cells (thus most of the dry snow zone) to build a full firn layer (i.e. until ice density is reached). The authors should thus support their statement (lines 226-227) that the profiles "averaged from the final 100 years of simulation results" (thus starting the averaging only after 160 years) are "steady-state density profiles". For example, after 160 years of an accumulation rate of 0.07 m w.e. $yr^{-1}$ (the limit assumed for warm dry snow zones in Section 3.2), only 11 m w.e. have accumulated, which corresponds roughly to a firn column of 20 m. I doubt that this represents a steady state. As shown in Figure 4 (T=-34°C), firn that is 100 years old is only at 600 kg $m-3$ density, showing that the firn layer is most likely not in steady state after 160 years and not even after 260 years. Concerning the fresh snow density, the A'76 model calculates surface densities by itself, while vK'17 and vK'17+ use a fresh snow parameterisation (not given in the manuscript...). But how is fresh snow density calculated for the A'10 model? The prognostic equation for $r_e$ should also be given or referred to.*

We are aware of the difficulties pertaining to fully spinning up the firn column in low accumulation grid-cells that represent most of the Antarctic Ice Sheet (AIS). That is why we only evaluate firn columns that exceed a thickness of 60 m. By filtering out columns that have not reached this threshold, we are only evaluating relatively high accumulation grid-cells mostly

representing the GrIS. This is also why we only show relatively warm grid-cells in Figs. 2, 3, and 4, i.e., because most of the grid-cells that have relatively low accumulation rates are much colder and have not reached a depth of 60 m.

To address concerns raised above, we are adding a demonstration showing the time evolution of the simulated firn density profiles used for evaluation. Because the deep density trends over the last 100 years are statistically indistinguishable from zero, with variations occurring almost exclusively in the top 10 m, our 260 year spin up is adequate for evaluating grid-cells that accumulate at least 0.2 m SWE $yr^{-1}$, covering most of the GrIS and parts of the AIS.

We will also add to the revision that the "A'76" model configuration uses a constant new snow density of 50 kg $m^{-3}$. And we will add the new snow density parameterization by van Kampenhout et al. (2017) to the revised Section 3, where we provide a streamlined introduction of each of the ELM experiments evaluated in Section 4. The prognostic equation for $r_e$ is included in the Snow, Ice, and Aerosol Radiative Model (SNICAR) referenced in lines 75-77. We will add $r_e$ after "ice effective grain size (from SNICAR)" for clarity.

*The approximation of vertical strain rates (line 244) is also unclear. This raises the same questions that I mentioned above about the dynamic variables when assuming a steady state. In my view, the authors should include a detailed explanation about how the steady state strain rates of A'76, A'10, vK'17 and vK'17+ are calculated. This holds for both the calibration step as for the ELM simulations (i.e. the values appearing in Figure 4).*

We agree that it can be difficult to follow the nuances associated with all the different model configurations, including both within the statistical model and the ELM experiments. However, all of strain rates calculated in our study are from the equations provided in Section 2.

As we attempt to mitigate this confusion, we are following your suggestion to expand the description of the statistical model. We hope our revision of Section 3 will reduce obscurity and, in addition to adding details discussed in previous responses, provide readers with clearer steps on how to reproduce our statistical modeling results and approximated strain rates from ELM simulations.

*When analysing and evaluating the results, there is a severe lack of quantification. This holds for both the Equilibrium climate simulations and the Twentieth century climate simulations. I give some examples:*

- *"the semi-empirical model improves the density profile" (line 260): improves with respect to what (I guess that the authors mean A'10 improves the density profile with respect to A'76)? And the statement of "improvement" should not be based on a mere visual comparison of Figures 2 and 3. Moreover, it should be clarified that the authors evaluate the model results against results from HL, which is not a guarantee of model accuracy.*

  You guess correctly, though you should not have had to guess at all.

  We quantify improvements with our new analysis, which computes RMSEs of ELM density profiles with reference to the nearest measurements from the SUMup dataset (Koenig and Montgomery, 2019). This new analysis supports our previous qualitative claims, which will be moved into a new "Discussion" section (5). We hope that our new format, having two separate "Results" and "Discussion" sections, will allow readers to better interpret our quantitative results without confounding influences from our more

speculative remarks.

- *"Densification tapers-off at lower densities (around 450 kg m$^{-3}$) for colder climates, a temperature-dependent effect enhanced with the model from Arthern et al. (2010)."* (lines248-250): from Figures 3 and 4, it is not obvious that this effect is [s]tronger in A'10 than in vK'17+. The enhancement of the effect should thus be quantified.
  Agreed.
  We will quantify this statement, move it into the discussion, and further qualify it by replacing "...effect enhanced..." with "...effect slightly enhanced..."

- *"A lower model variance occurs when it does not covary with the empirical model. This effectively reduces a model's prediction risk if it does not also result in an increased bias."* (lines 303-304): If the authors discuss about the bias of models, they can simply add a "Bias" column in Table 2.

- *"both models show improvement compared to their original counterparts (ELM v1 and CLM)."* (lines 337-338): this is impossible to evaluate for the reader because (1) only the results of A'10 and vK'17+ are shown in Figure 6, and not the ones of their so-called "original counterparts" (which are A'76 and vK'17 I suppose), and (2) there is no quantitative evaluation of the models' performance with respect to the observed data (e.g. RMSE, bias, etc.).
  Our lack of a sufficient quantitative analysis is the fundamental motif in the open discussion of this study.
  Our revision includes an expanded Section 4 that streamlines the presentation of ELM experimental results and also includes RMSEs computed with reference to the SUMup measurement dataset (Koenig and Montgomery, 2019).

- *"Encouragingly, our simulation results compare well with firn density measurements and indicate an improved capability in the ELM."* (lines 348-349): same remarks as for the previous point.
  Again (not surprisingly), our lack of a sufficient quantitative analysis is the fundamental motif in the open discussion of this study.
  In our revision we expand Section 4, which now streamlines the presentation of ELM experimental results and also includes RMSEs computed with reference to the SUMup measurement dataset (Koenig and Montgomery, 2019).

3. *The novelty and objective of the study*
   *GMD review criterion:*

   - *Does the paper present novel concepts, ideas, tools, or data?*

*Firn model optimisation has been addressed in numerous studies over the recent years (e.g. Ligtenberg et al., 2011; Kuipers Munneke et al., 2015; van Kampenhout et al., 2017; Smith et al., 2020; Verjans et al., 2020). Four of the studies mentioned have already investigated the optimisation of the model of Arthern et al. (2010), for Antarctica, Greenland or both. An easy and straightforward way to improve the ELM would be to implement the parameterisations developed in these studies. If the authors want to address the same problem, they should propose a new, original method. However, in contrast to the existing literature, they do not calibrate the model of Arthern et al. (2010) with observations but with HL-computed strain*

*rates. And, as mentioned above, it is unclear to me how they calibrate a dynamic model to steady state strain rates. They should justify why their methodology is better suited to their objective than using what other researchers have already accomplished. Moreover, the objective stated in the conclusion of improving the capability of the ELM to better simulate refreezing rates in firn is not in line with the study itself. The focus of the calibration is on dry firn densification and does not support the statements at lines (377-381): "With an evaluation of the simulation of dry firn densification, we have optimized the ELM firn model for future studies of the impacts of liquid water on firn density and SMB. Ultimately, this study seeks to enable better predictions of SLR as a direct result of surface melt and mass loss from the GrIS."*

Currently, as far as we know, there is only one other ESM that attempts to include snowpack processes at this level of detail (CESM), which we discuss and compare to in our study. We are not aware of any other firn parameter optimization studies that provide a complete set of values that can simply plug into ELM. Furthermore, because ELM includes two dry snow densification processes (destructive metamorphism and overburden pressure compaction), assimilating published firn densification models into ELM requires calibrating their coefficients to account for both compaction terms. What is also novel about our study is that we experiment with combinations of existing snowpack and firn densification parameterizations. We optimize using the model of HL'80 as calibration data because it can more easily be applied to regions where observations are limited. Even with a complete observational data product, density profiles would still need a smoothing filter to improve the numerical stability of the vertical differentials used in the calculation of strain rates. We use this method to optimize strain rate data to steady strain rates, though we failed to provide adequate documentation regarding the pressure, temperature, and grain size inputs to the model.

We aim to address this problem by providing more details regarding the statistical firn model used to calculate pressure and temperature profiles. We are also removing the misleading quote stated above. Our revised manuscript will also include some additional discussion in the introduction to stress that the goal here is not to make the best firn model ever and stick that in our ESM. Rather, we need a model that balances physical realism against computational cost.

*If the authors do want to better capture liquid water effects, they should focus on this very challenging topic by studying the mechanisms of wet firn compaction, meltwater percolation and refreezing.*

Wet firn densification is outside the scope of this study. It could be the topic of a future study, but it would be premature to try to improve the wet processes until the dry processes have been fully vetted.

4. *The clarity of the manuscript*
   *GMD review criterion:*

   • *Is the overall presentation well structured and clear?*

   *It is very difficult for the reader to understand the different steps of the study. The authors alternate between different ways to refer to a same thing. For example, the A'10 model is sometimes referred to as "A'10" and sometimes as "the semi-empirical model", the vK'17 model is sometimes referred to as "vK'17", sometimes as "the CLM" and sometimes as the "Snowpack model" (see Figure 5). Similarly, it is never clearly stated that the "empirical*

*strain rates" correspond to the ones computed with the HL. A first, simple way to improve the clarity would be to consistently use the terms A'76, A'10, vK'17 and vK'17+ throughout the manuscript, including in the captions of the Figures. In line with this, the Section 3.1 should be split in four subsections that clearly detail each of these four models instead of subsections presenting equations which are subsequently assigned to the models in a confusing way.*
We appreciate these constructive comments.
As a first step, we are following your suggestion regarding Section 3.1. Second, we will remain consistent in our labeling of ELM experiments throughout the manuscript, which entails replacing the implicit descriptors you mention above.

*The Figures and Tables also lack clarity. In Figures 2 and 3, why are high values of accumulation only shown at depths greater than 15 m? Even in a high-accumulation climate, there will always be a shallow and a deep part of the firn column. And how were the surface density values chosen for the HL-computed profiles? The caption of Figure 1 mentions that the new firn model "can extend as deep as 80 m", whereas it is always presented as a "semi-infinite" grid in the text. Which of these two statements is true? In Figure 4, why are there points without a vertical error bar? And why do some points have a horizontal error bar (age should be well-determined for any firn layer of any model run)? In Table 1, equation numbers could be provided for each model to know which equations apply to which model. In Table 2, the model names should be used in the column "Densification model" instead of the mechanisms applied and the variable for which the statistics are calculated should be specified in the caption (presumably strain rate values). Improving the structure of the manuscript could possibly help decrease the degree of confusion for the reader when trying to understand the study.*

The empirical model from Herron and Langway (1980) gives density profiles that do not vary with accumulation rate for depths less (more shallow) than the critical depth, i.e., where density is less than 550 kg m$^{-3}$. In Figs. 2 and 3, the high values of accumulation are obscured by the lowest value. The surface densities range from 300 to 380 kg m$^{-3}$, which are appropriate for ice sheets (van Kampenhout et al., 2017; Herron and Langway, 1980).

The maximum depth of the new snowpack model is a recurrent point of confusion for readers. While the new vertical grid has a bottom most layer without a bound on its maximum thickness, we choose 60 m as the limit for most of our analysis because that is roughly how far below the surface the 15 layers of finite thickness will extend when they reach their maximum thicknesses (Figure 1). Therefore, the grid is truly semi-infinite, and we choose to exclude the semi-infinite layer from (and thus restrict the vertical extent of) our analysis because the model cannot resolve dynamic variables deeper than 60 m at a vertical scale appropriate for simulating densification (see also response to anonymous referee 2).

In Fig. 4, it appears that some of the vertical error bars are obscured. We speculate this is caused by poor rending of vector graphics. Horizontal error bars represent grid-cell standard deviations grouped by their Eulerian reference layer (i.e. fixed depth). A layer $z_i$, e.g., across the global land surface domain represents firn that varies significantly in age. This results in a blurred analysis that we account for with horizontal error bars.

To improve clarity, we changed the wording in the caption of Fig. 1, which replaces "80 m" with "semi-infinite." Next, we will try using a different image format (or change the line styles) for Fig. 4 to elucidate the vertical error bars. The revision will more clearly explain the meanings of all the symbols (axes, limits, etc...) in the caption. Finally we appreciate

your suggestions on how to clarify information given in Tables 1 and 2 and will follow them accordingly.

- *line2:*
  *Change "consist" to consists.*
  Corrected.

- *line 15:*
  *I doubt that any paleoclimate study uses Earth System Models to determine pore close off depth and timing.*
  They are not able to using current Earth System Models, further motivating this study.

- *line 25:*
  *Repetition of "coupled".*
  We will rephrase this sentence in our revision.

- *lines 32-35:*
  *As far as I understand, there is a contradiction between "does not yet exist" and explaining the implementation of the advanced firn model in the CLM.*
  The advanced firn model in CLM (version 5) limits its snow water equivalent (SWE) depth to 10 m, which likely corresponds to a maximum allotted firn thickness of less than 30 m. Furthermore, the CLM firn densification model is not valid for densities greater than 550 kg m$^{-3}$ (stage 2).
  We will elaborate this point in our revised "Discussion" section (5).

- *lines 47-48:*
  *Change "those predicted by Herron and Langway (1980)" to "those predicted by the model of Herron and Langway (1980)." Moreover, I suggest using a consistent way to refer to this model (e.g. HL'80).*
  Thank you; we will follow these suggestions.

- *lines 52-55:*
  *Add an explanatory sentence about the fact that snowpack models and firn models also have a different vertical scale of application.*
  Nice suggestion. We will do so accordingly.

- *Section 2.1:*
  *Provide units of all the variables and quantities presented. This will make clear that there are some unit inconsistencies in some of the equations (which I give below).*
  We will check all equations and variables for consistency and add units accordingly.

- *Equations 1, 2, 3 and 4:*
  *The variables $\dot{\epsilon}$ and $\left( \frac{1}{\Delta z} \frac{\partial \Delta z}{\partial t} \right)$ are equivalent to each other as far as I understand. Use either one of the two notations.*
  We will add a statement of their equivalence and refer to their following quantities as $\dot{\epsilon}$.

- *line 78:*
  *Specify if $\left( \frac{1}{\Delta z} \frac{\partial \Delta z}{\partial t} \right)_{dm}$ is also considered in CLM (v5).*
  Thank you for pointing this out. We will clarify that the destructive metamorphism (dm) parameterization is also used in CLM (v5).

- *line 82:*
  *In CLM(v5), $c_\eta = 358$ kg m$^{-3}$ and $f_2$ was set to 4. Only $f_1$ accounts for the effects of liquid water and not $c_\eta/(f_1 f_2)$.*
  In CLM(v5), $c_\eta = 450$ kg m$^{-3}$
  (https://github.com/ESCOMP/CTSM/blob/master/src/biogeophys/SnowHydrologyMod.F90,
  line 3766). This conflicts with van Kampenhout et al. (2017), who specify $c_\eta = 358$ kg m$^{-3}$.
  Because we adopt the same constant value for $f_2$, the quotient $c_\eta/f_2$ can be lumped together
  as a constant coefficient, and the entire expression $c_\eta/(f_1 f_2)$ still depends on the liquid water
  content via $f_1$.

- *line 89:*
  *The characteristic depth is not "a single valued proxy for a given site's full density profile"
  but only for the upper density profile.*
  We will replace "full" with "upper."

- line 89:
  *No s at "stages."*
  Thank you. We will correct.

- *line 89:*
  *Add here the explanatory sentence about why models assume a two-stage densification process.*
  Done.

- *line 92:*
  *The sentence "Empirical firn densification models typically employ analytic functions that
  assume a steady-state density profile" is not true. Only the model of Herron and Langway
  (1980) and a few others provide analytic functions but almost all of the recent firn models
  are dynamic models.*
  It is our understanding that the older "empirical" models use statistical methods (linear regression) to calibrate analytical formulations directly to density measurements.
  We will replace the above sentence with "Empirical firn densification models have historically employed analytic functions that assume a steady-state density profile."

- *line 93:*
  *Rephrase.*
  Will consider.

- *Equation 5:*
  *This equation is erroneous. The units of the left- and right-hand sides do not match. The
  correction is: $w(z) = \frac{A}{\rho(z)/\rho_w}$, where $\rho_w$ is the density of water (1000 kg m$^{-3}$).*
  Here is our dimensional analysis:
  $w(z)$ [m s$^{-1}$] = $A/\rho(z)$ [kg m$^{-2}$ s$^{-1}\times$ kg$^{-1}$ m$^3$] = [m s$^{-1}$], where $A$ is the accumulation rate
  in terms of kg m$^{-2}$ s$^{-1}$ equivalent to mm of snow water equivalent (SWE) per second.
  We will correct line 100, replacing "(... SWE per year)" with "(... mm SWE s$^{-1}$)."

- *Equation 7:*
  *Again, this equation is erroneous and there is a unit inconsistency. The correction suggested
  above, fixes the error.*

Our correction above fixes the error.

- *Equation 8:*
  *The variable $P$ is defined here as the "overburden pressure", which makes it equivalent to $\sigma$. I suspect that this variable corresponds to the $P$ as defined in Equation 9, which should be called the "grain-load stress." I underline here that in the model of Arthern et al. (2010), it is really $\sigma$ that is used and not the grain-load stress. The authors should explain why they differ from the original model of Arthern et al. (2010) on this point.*

  These are some interesting points that are confusing in the literature. Here, we reserve $\sigma$ for the vertically-integrated (column) areal density [kg m$^{-2}$] and $P$ for the overburden stress [Pa]. Because of inconsistent use of tuning coefficients in the literature, this subtle distinction is only apparent in dimensional analysis. In CLM(v5), for example, van Kampenhout et al. (2017) are indeed referring to $\sigma$, which requires being multiplied by the acceleration due to gravity ($g$) [m s$^{-2}$] to convert this quantity to "overburden pressure" as a stress. These conversions get washed out by tuning coefficients, which are (again) inconsistent across the literature.

  In line 158, we do refer to the "grain-load stress" as you suggest. You are correct, however, in that Arthern et al. (2010) are referring to the "overburden pressure" in their model and not the "grain-load stress" as we adopt in our implementation of their model. Our choice is motivated in the discussion of Arthern et al. (2010), where they state that their model results in densification that occurs too slowly over the interior ice sheets. Therefore, we apply the "grain-load stress" instead of "overburden pressure," as in eq. (9), because that increases the magnitude of their modeled strain rates by approximately a factor of 2. Furthermore, after applying our offline calibration method, the distinction between these two approaches (neither of which represents a mechanically sound method, by the way), is of second order importance.

  For clarity, we will add a brief discussion (similar to above) after eq. (9) motivating our choice of "grain-load stress" $P$.

- *lines 176-177:*
  *Specify that the "plausible firn density-versus-depth profiles" were computed with the different models and the HL.*

  We already mention (albeit compactly) that the "plausible firn density-versus-depth profiles" were computed using the model of Herron and Langway (1980) in lines 182-183.

- *Section 3.2:*
  *Why do the authors decide to draw annual temperatures from a distribution representative of the global Earth climates instead of the polar climates? Is the objective to have much more values close to $T =$-25°C? This should be clarified.*

  Because we will soon study what happens when dry firn starts to melt, we optimize our calibration routine to be more representative of values close to $T =$-25°C. In the future, as discussed above, we could drive the statistical model with input data from a regional climate model or polar reanalysis, however, this is beyond the scope of our present study.

  In our revision, we will specify our motivation for favoring values close to $T =$-25°C.

- *How are all these values decided:*
  We reviewed the work by Herron and Langway (1980) and arrived at the following values

based on the domain of observations applied in their study.

    – *-25°C as a threshold (whereas ELM simulations involve warmer sites)*
    This is a rough estimate partitioning Greenland's dry snow zones that we obtained from Cuffey and Paterson (2010).
    We will add the reference.

    – *-51°C as limit between low- and high-accumulation sites (many sites can have $T > -51°C$ and $A < 0.07$ m SWE $yr^{-1}$)*
    Because we did not want unrealistically high accumulation values for $T < -51°C$, we selected this limit based on observations tabulated by Herron and Langway (1980). Our statistical model does not consider low accumulation values for $T > -51°C$ because they are not well represented by the model of Herron and Langway (1980).

    – *surface density values between 300 and 380 kg $m^{-3}$*
    These limits were deduced from the range of values selected by Herron and Langway (1980) most applicable to our domain of interest (i.e., dry snow zones).

- *line 196/*
  *Specify the resolution of "coarse-resolution".*
  The horizontal resolution of the ne11 grid is 2.8°.
  As requested, we will provide the nominal resolution when referring to ne11.

- *line 197:*
  *What does "(an "I-compset" at ne11 resolution)" mean?*
  This is our Earth system modeling jargon, meaning stand-alone land surface model (i.e. un-coupled ELM) at 2.8° horizontal resolution.
  We will add the ne11 nominal resolution.

- *line 199:*
  *Change "January 1st" to "January 1st 1901."*
  Okay.

- *line 206:*
  *Define "restart runs". In general, it is good practice to define any term used that may not be straightforward to everyone reading the study.*
  "Restart runs" are in parentheses. They are defined in the corresponding sentence.

- *Table 1: Add a column for $\rho_{dm}$ values. Add another column that indicates which equations apply to which model, with the corresponding equation numbers (see Major Comment 4).*
  Thank you for the helpful suggestion. We will update Table 1 accordingly.

- *lines 209-215:*
  *All the details provided about the observational data should be given in a separate sub-section. Are the sites of measurements affected by surface melt? And are these firn core measurements open access?*
  We will move this paragraph into its own subsection and offer our perspectives on the applicability and quality of these publicly available measurements.

- *Section 4:*
  *There are a lot of speculative statements in this section. I suggest splitting it into a section*

*Results and a section Discussion, so that the reader can distinguish between model results and the thoughts of the authors.*

Thank you for this excellent suggestion. We appreciate your comment as it is clear you have put some thought into how the manuscript can be better structured to benefit readers.

We are adding a separate "Discussion" section (5). This entails moving most of the material currently in Section 4 into the new Section 5 and adding an updated analysis to the beginning of Section 4. We think you will appreciate seeing your suggestions realized in our revision, which will resolve many of the issues you bring to our attention.

- *line 219:*

  *Change "accumulations" to "accumulation rates."*

  Done.

- *lines 217-222:*

  *Here, the entire methodology is again defined. This can be confusing for the readers. For example, I suggest rephrasing the sentence "To improve the accuracy of our firn model simulations, we optimize compaction terms against empirical strain rates using statistical modeling" because this was the point of a previous section.*

  We disagree that the "entire methodology is again defined." Rather, this is a very brief summary of the methodology and introduces our analysis below.

  This paragraph will be trimmed and moved in our revision.

- *line 229:*

  *Does the statement "the mean annual temperature is within a couple degrees of -25 °C" refer to the results of Figure 2 for T=-27°C? If so, it would be clearer to give the exact mean annual temperature value.*

  We will give the exact values of the mean annual temperatures instead of the imprecise statement above.

- *lines 236-237:*

  *"These simulations demonstrate a stronger effect of temperature on densification rates, resulting in more variation in density with depth (Fig 3)." I think that this sentence means more variable depth-density profiles according to the mean annual temperature. If so, it should be rephrased.*

  This sentence is being moved into and rephrased in our new "Discussion" section.

- *lines 242-243:*

  *Specify which models use the "better fresh snow density parameterization".*

  We will add discussion of which models result in better fresh snow density.

- *lines 260-261:*

  *Strange use of commas.*

  We removed the comma after "ELM".

- *line 264:*

  *What is "over-densification"?*

  Unrealistically high densification rates.

  Here, and throughout the manuscript as necessary, we will refrain from using this term in favor of a better description.

- *line 265:*
  *The notion of "density profiles that vary too weakly with depth" should be replaced with the one of density that increases too weakly in depth.*
  We will make the relevant changes.

- *line 265:*
  *What does "Their" refer to?*
  Arthern et al. (2010). We will specify.

- *lines 274-277:*
  *Are these the only coefficients included in the optimisation? Or the only ones that were decided to be changed? See Major Comment 1.*
  Yes, these are the only coefficients included in the optimization, as we describe in our response to your first major comment.
  We will clearly state these details in our expanded statistical modeling subsection.

- *line 276:*
  *The coefficient $f_2$ is related to the grain radius (see Vionnet et al., 2012). If the ELM calculates grain size, $f_2$ can be calculated accordingly. It is crucial that the authors clarify why they decide not to follow the original formulation of $f_2$ (as a function of grain size) but to consider it as a pure tuning factor. This is all the more relevant since it is emphasised throughout the manuscript that firn models need to account for microphysical features.*
  Here, we follow van Kampenhout et al. (2017). But perhaps we should experiment with the original formulation from Vionnet et al. (2012) in the future.
  For now, we will add a discussion regarding this parameter, including its original value, the value adopted by CLM(v5), and in our study as a pure tuning factor.

- *line 286:*
  *Specify that the density model coefficients are calibrated to the HL-computed strain rates.*
  We will specify this detail.

- *lines 288-290:*
  *Specify the variable for which the statistics are calculated. See Major Comment 4.*
  This sentence refers to the strain rate $\dot{\epsilon}$, which we will specify.

- *lines 293-298:*
  *How do the authors explain these results?*
  These results suggest that the overburden pressure dependence, which is part of most firn densfication models, complicates its calculation of realistic strain rates. A simpler, temperature-density dependent equation – while originally used for calculating compaction due to destructive metamorphism – can be modified so that it results in densification rates more highly correlated with those predicted by the model of Herron and Langway (1980). This is a surprising result that challenges the use overburden pressure in firn densification models that depend mostly on their functions of density.

- *Table 2:*
  *The mention to eq. (5) is an error because it is not the one for destructive metamorphism and $c_5$ does not appear in this equation.*
  Thank you for bringing this to our attention.

We will replace "eq. (5)" with "eq. (2)", which is the correct equation.

- *line 300:*
  *The value of $R^2 =0.67$ is valid only for strain rates in the second stage. Note also that Table 2 shows $R^2 =0.66$.*
  We will add that $R^2 =0.50$ for the first stage.
  A difference in $R^2$ of 0.01 is negligble and a result of repeating experiments that generate input values at random.

- *line 303:*
  *"A lower model variance occurs when it does not covary with the empirical model". This statement sounds like a general statement, but I believe that it is applicable only to the results of this specific study.*
  We will rephrase as "These smaller variances indicate that our experimental model does not covary with values derived from the model of Herron and Langway (1980)"

- *line 306:*
  *What does "these results" refer to? The paragraph above is about variances in the compaction rates.*
  Will change "these results indicate" to "our covariance analysis shows"

- *lines 306-307:*
  *"negative correlations between overburden pressure and empirical strain rates": this is explained by higher overburden being applied to deeper firn, which is at higher density and thus compacts less. A simple explanation could be provided to the reader.*

- *lines 311-312:*
  *Note that Equations 3, 4 and 8 are directly dependent on density.*

- *lines 320-328:*
  *Are these results or speculations? Where do these values come from? And did the "statistical computing" focus only on these specific coefficients of A'10? Such conclusions should not be stated without quantitative results to support them. I emphasise again the need to clarify the optimisation method and its results.*
  These are results from our statistical computing, which we speculate will improve the A'10 experimental results. We will include results from this ELM experiment ("A'10+") in our revision, thus reducing the speculative nature of these results.
  We are reorganizing and expanding Section 3 so that our statistical model is more transparent, expanding Section 4 so our ELM experimental results are more streamlined, and adding a "Disucssion" section (5) to signal a clear distinction of results from our more speculative remarks regarding future model development.

- *lines 331-332:*
  *Why did the authors decide to use the optimised vK'17 but not the optimised model of A'10?*
  We had not completed ELM experiments using the optimized model of A'10 ("A'10+").
  With the A'10+ ELM experiment now complete, we will provide these results in our revised Section 4.

- *line 333:*
  *Typo "the the"*

Corrected.

- *line 341:*
  *"ne11" is not defined.*
  The horizontal resolution of the ne11 grid is 2.8°.
  As requested, we will provide the nominal resolution when referring to ne11.

- *line 350:*
  *The statement "we should focus on the near surface layers, as they contain the primary SMB components" should be clarified.*
  We removed this sentence.

- *line 351:*
  *"it could be necessary to model the upper most 20 m": Is this figure of 20 m supported by the results? If not, references should be provided.*
  We removed this sentence.

- *lines 352-353:*
  *"the optimized version (vK'17+) is likely the better choice for implementation into the next major release of the E3SM": Again, it is unclear if this is speculative or if the results provide strong evidence for this. And how would an optimised version of A'10 compare to vK'17+?*
  With ELM simulations of an optimized version of A'10 now complete, we can offer a more thorough discussion of the optimized model comparisons. In our revision, we include results from both of these particular ELM experiments in the expanded Section 4 and their respective RMSEs calculated with reference to the SUMup dataset (Koenig and Montgomery, 2019). Furthermore, the comparison of A'10+ and vK'17+ model accuracy is the focus of a paragraph in our revised "Discussion" section.

- *lines 353-355:*
  *A low bias is usually preferable to a high bias. What is meant by this sentence?*
  We removed this poorly worded sentence.

- *line 364:*
  *"In the near future, this could be the entire GrIS": What is the "near future"? And references should be provided.*
  We will replace "In the near future" with "By 2100" and cite Machguth et al. (2016) as recent evidence suggesting that feedback processes will likely cause the GrIS melt extent to expand rapidly in a warming climate.

- *Conclusions:*
  *See Major Comment 3.*
  Regarding lines 377-381, we replace "... have optimized.... Ultimately, this study seeks to enable better predictions of SLR as a direct result of surface melt and mass loss from the GrIS." with "...prepared.... This study marks progress toward better predictions of SLR caused by surface melting of the GrIS."

- *line 370:*
  *Change "with steady-state empirical models" to "with a steady-state empirical model."*
  Thank you; we will change the text accordingly.

- *lines 370-371:*

*I do not believe that the analysis is "similar to that by van Kampenhout et al. (2017) for CLM."*

We will replace "analysis" with "model development", which parallels the work of van Kampenhout et al. (2017) in CLM.

- *Code and data availability:*
*The availability of the firn core measurements should be provided in this section.*
We will seek guidance from the topical editor regarding the policy associated with previously published data.

- *Figures:*
*See Major Comment 4. Also, in all Figure captions and legends, the models should be consistently referred to as A'76, A'10, vK'17 and vK'17+.*
Please see our response to Major Comment 4.

- *Figure 1:*
*The vertical extension of "80 m deep" (caption) contradicts the "semi-infinite" stated in the text.*
We changed the wording in the caption to provide clarity on the semi-infinite bottom layer and removed the mention of 80 m.

- *Figures 2 and 3:*
*The vertical/horizontal lines at $\rho = 550$ kg m$^{-3}$ and $\rho = 550$ kg m$^{-3}$ can be removed to improve the clarity of the Figures.*
*Why do high accumulation rate values appear only at great depth?*
See our response to Major Comment 4.

- *Figure 4:*
*"for various plausible accumulation rates (0.11–0.50 m SWE yr.$^{-1}$ )": Why is 0.11 m SWE yr.$^{-1}$ chosen as lower limit?*
This limit was chosen as a suitable value based on observations referred to by Herron and Langway (1980).
*The construction of the horizontal error bars is not clear to me.*
We described the horizontal error bar above, but will clarify this in a revised figure.

- *Figure 5:*
*Do these plots show the results of A'10 (top) and vK'17 (bottom)?*
"A'10" and "vK'17" refer to implementations in ELM. Fig. 5 shows results from the combined destructive metamorphism plus overburden pressure compaction equations after calibration.
The revised structure will clarify the distinction between offline firn model configurations targeted in the optimization versus similar implementations in ELM experiments.

For a direct look at how we address these comments, please see the forthcoming revised manuscript.

Sincerely,

Adam M. Schneider et al.

---

## Author Comment (AC2) · 13 Nov 2020

13 Nov 2020

re: "Review of Schneider et al.", Anonymous Referee # 2, 11 Sep 2020

We appreciate your positive review and are considering your comments as we revise the manuscript. Following below is our response (in blue) to your general and specific comments, which are italicized here for reference:

- *This is a very well written manuscript that is motivated and articulated clearly. I am not an expert in ESMs or their various component models, and therefore cannot comment on the authors' specific implementation of firn compaction models into the E3SM Land Model.*
  Thank you for the kind remarks. We appreciate your perspective and are working to reach readers with and without established expertise in Earth system modeling.

- *The stated goal of this work is to more accurately simulate snowpack evolution in the ELM, including over the Greenland and Antarctic ice sheets. With this goal, why do the authors increase the snowpack from 1m (in the previous version) to up to 60m in this version? In the dry snow zone, it's common for firn column to vary in depth from 50 to 120m (Cuffey & Paterson, 2010), depending on site conditions. Many sites have firn columns deeper than 60 m (and even the 80 m depth that Figure 1 indicates is possible), and therefore this estimate of snowpack evolution won't be valid for large swaths of the Greenland and Antarctic ice sheets. If instead the authors are referring to depths in m SWE, this should be made explicit within the text and in the figure axes.*
  The maximum depth of the new snowpack model – the distance from the surface, not in terms of snow water equivalent (SWE) – is a recurrent point of confusion for readers. While the new vertical grid has a bottom most layer without a bound on its maximum thickness, we choose 60 m because that is roughly how far below the surface the 15 layers of finite thickness will extend when they reach their maximum thicknesses (Figure 1). Therefore, the grid is truly semi-infinite, but we choose to exclude the semi-infinite layer from (and thus restrict the vertical extent of) our analysis because the model cannot resolve dynamic variables deeper than 60 m at a vertical scale appropriate for simulating densification. Because the focus of our study is on getting the bulk density structure of the firn correct, our primary domain of interest is where most of the densification takes place, i.e., closer to the surface. By nature, the representation of firn in an Earth system model has to have some approximations and limits in its accuracy and detail. For our purposes, we deem an evaluation of the top 60 m adequate for capturing the vast majority of the firn densification on ice sheets. Much lower than 60 m, densification rates become relatively small and are of second order importance. To alleviate confusion, we changed the wording in the caption of Fig. 1 to better reflect our semi-infinite snowpack grid.

- *In various places the authors describe "improvements" or "slight improvements" between models, especially in Seciton 4.1. How do the authors determine these improvements? Are they quantifiable?*
  The qualitative improvements we describe are mostly justified by directly comparing density profiles and estimated strain rates simulated in ELM to the empirical model of Herron and

Langway (1980). We attempt to show these comparisons in Figs. 2, 3 and 4, but due to our presentation of the results and discussion, it can be difficult for readers to evaluate data from all the experiments. It is also clear that claims of improved density profiles need further quantification.

To improve the readability and interpretation of our main findings, we are splitting Section 4 into two separate "Resuts" and "Discussion" sections. In these sections, we first present density profiles, now separated by ELM grid-cells, from each experiment in the order which they are described in our revised Section 3. To quantify improvements, we add to Section 4 root mean squared errors (RMSE) calculated with reference to a comprehensive dataset of available measurements described by Montgomery et al. (2018). We also replace "improvements" with model "developments" unless justified with quantitative analysis.

- *Section 3.1 – The abstract states that the authors improve the depth of snowpack in the ELM from 1 m to up to 60 m, while the Figure 1 caption states that the new model layers can extend to 80 m. Why is there a discrepancy between these two extended depths?*

  Our version of ELM now has a vertical resolution of roughly 8 m in the top 60 m of firn. With a semi-infinite bottom most layer, ELM can also accommodate a total firn thickness much deeper than 60 m, as demonstrated in Figure 1. Because the grid spacing below about 60 m deep increases indefinitely, we generally exclude results from this semi-infinite layer when its depth exceeds 60 m, hence the 60 m value stated in the abstract.

  Again, to better reflect this important detail, we changed the wording in the caption of Fig. 1, which now includes "semi-infinite."

- *Section 3.3.2 – a brief description of the Greenland sites used from the Mosley-Thompson et al. (2001) study would be helpful here. There were quite a few cores in that thorough study. How did the authors decide which cores to average into a composite GrIS density profile? The accumulation rate varies quite a bit across Greenland, especially north to south. Were northern and southern sites averaged together? Additionally, only sites near Siple Dome were used in Antarctica. Since sites in East Antarctica has much lower mean annual temperatures and accumulation rates that Siple Dome, it may be more appropriate to claim that this applies to West Antarctica than both Antarctic ice sheets.*

  We agree that the comparison against the Mosley-Thompson et al. (2001) study is underdeveloped and over simplified. It is also insufficient to extrapolate results from our Siple Dome case study to the entire Antarctic Ice Sheet(s).

  To improve (and update) these case studies, we are adding to Section 4 a new analysis that controls results for geographic location and includes RMSEs calculated with reference to a comprehensive set of available measurements described by Montgomery et al. (2018). This new analysis, in addition to providing geographically controlled metrics for evaluating model performance, better serves our study's primary objective of improving the representation of dry firn densification in ELM's accumulation zones.

- *Section 4.1 – Lines 237-239: It would be helpful to the reader if the authors indicate which model they're referring here when they say "...this dynamic implementation of eq. (8)..." and "...the original compaction parameterization, from eq. (3)." Are these referring to models vK'17+ and A'76, respectively?*

  Another reviewer also brought up that Section 4 is confusing and that we should revise Section 3 to improve the clarity of the manuscript.

To minimize guess work a reader must endure, we are following Vincent Verjan's suggestion of reorganizing Section 3 so that its (sub-)subsections will correspond to and identify the unique model configurations that appear throughout the manuscript. Our revised Section 4.1 also follows that outline.

- *Lines 225-227: What is the justification for using the 1901-1920 reanalysis data to generate the steady-state density profiles? Is this considered an average period of time? If so, what metric was used to determine that it was best to use the data from 1901-1920?*
  We use the 1901-1920 time period for two reasons. First, according to the IPCC Fifth Assessment Report, these decades represent a relatively stable surface climate in terms of the global mean temperature (Hartmann et al., 2013). Second, these decades are the earliest decades available from the CRUNCEP atmospheric forcing data, which brings us as close to a pre-industrial climate forcing as possible.
  In our revision, we will add this motivation where we introduce the atmospheric forcing (previously Section 3.1.1.).

- *Lines 225-227: The spin-up of the model simulated 260 years of snow accumulation is adequate for creating a typical dry snow-zone firn column in Greenland but would not reach back far enough to erase the natural firn density profiles in East Antarctica. Why was a spin-up of 260 years chosen?*
  We are aware of the difficulties associated with spinning up the snowpack and firn column for East Antarctica. Doing so would simply require computational resources beyond our allotment.
  We choose 260 years for the duration of the spin-up period because it achieves a balance of fulfilling a large number of grid-cells of interest while being computationally feasible. The exact value of 260 years is somewhat arbitrary, which we reached based on rough calculations for how long it would take to reach pore close off based on values from Cuffey & Paterson, 2010)

- *Lines 227-229: are the authors referring to two of the examples given (-27C and -20C) here? Describing results for scenarios with mean annual temperature "within a couple degrees of -25C" is imprecise and leaves the reader wondering if they're missing a panel in Figure 2 for the -25C scenario (and other scenarios in between).*
  We agree that the presentation of results here are imprecise and need further specification. To help readers better interpret these results, we are changing the panels in Figs. 2, 3, and 4 to show mean annual temperatures of -39, -32, and -25 $°C$, while modifying descriptions in the text to be more precise.

- *Lines 229-230: It'd be helpful to indicate here that the results from the Herron & Langway model are the colored bars in Figure 2.*
  That is already stated in the figure captions, but we will add it to the text to improve readability.

- *Lines 237-239: In the discussion of Figure 3 here, the authors describe that the dynamic implementation of eq. (8) in the ELM (Again, is this the vK'17?) results in characteristics depth (550 kg/m3) more consistent with Herron & Langway for T > -32C, but the authors do no[t] present any of the results for the T=-32C scenario. How was this cutoff determined? Additionally, the panel for the T = -20C scenario shows that neither A'10 nor vK'17+ do a*

*good job predicting the characteristic depth. Therefore, how do the authors conclude that the results are more consistent with H&L for T > -32C?*

It is (again) confusing for readers to understand which model configuration corresponds with which equations. This is due to the poor structure of Sections 3 and 4. The -32 °C "cutoff" is a rough one, which would be more apparent with a different (or more complete) set of panels to go with Figs. 2, 3, and 4. The characteristic depth is not well predicted for temperatures greater than -25 °C, where it is likely that melt starts accelerating densification in ELM, an effect not really accounted for by Herron and Langway (1980).

We are applying several changes to the manuscript to clear up these confusing matters in our revision. First, as described above, we are restructuring Sections 3 and 4 so that they will streamline the ELM experiments into (sub-)subsections, hopefully removing some guess work readers must currently endure to interpret the results. Second, we are changing the mean annual temperatures shown in Figs 2, 3, and 4 (i.e., to -39, -32, and -25 °$C$) to better reflect the specific objective of the study, which focuses on dry firn densification and thus cannot make definitive conclusions for regions that are warmer than roughly -25 °C. Third, we will clarify that our interpretations stated above do not pertain to warmer regions where melt probably affects densification rates and thus deteriorates the comparison validity against Herron and Langway (1980).

- *Figures 2 & 4: which lines are the authors referring to in these captions when they say "Line graphs show 100-year means from ELM simulations..."? All of the lines (except dashed & dotted?)? This vague statement is confusing due to the number of lines plotted.*

  We are referring to the ELM simulation results.

  Because this is not clear, instead of plotting multiple ELM experiments on the same figure, we will separate ELM results by experiment in Figs. 2, 3, and 4, which we are moving into a new, separate "Discussion" section to emphasize a new analysis in Section 4 that more directly evaluates ELM simulation results against observations.

- *Figure 2 – how are the range of surface densities used in the empirical modeling shown in the figure?*

  We choose 300 to 380 kg m$^{-3}$ for the range of surface densities to plug into the model of Herron and Langway (1980). These values, though somewhat arbitrary, are selected to be representative of surface densities observed on ice sheets.

- *Figure 3 – comparing the steady-state density profiles generated for the 3 mean annual temperatures to sites with similar conditions (Table 2.2, Cuffey & Paterson, 2010), it appears that the ELM and empirical modeling results reach pore close-off density at too shallow of a depth for -20C, empirical modeling results.*

  Yes. Densification does happen quickly for relatively warm regions. This might indicate that the models' temperature sensitivities are too strong, or that the effects of melt are muddying the comparison to what is generally an empirical model for dry firn densification.

  Because this study focuses on dry firn densification, we are removing all our results and interpretations from ELM grid cells that have mean annual temperatures greater than -25 °C. This includes in both Figs. (2, 3, and 4) and in text.

- *Figure 4 – The stated goal for this figure is to better understand what drives rapid densification near the surface. Therefore, it would help the reader to have a second x-axis displaying*

*the depth. Since we don't have a depth-age scale, it's hard to interpret where the near-surface is in this figure. What causes the jump in vertical strain rate in the empirical modeling results (colored lines)? Additionally, the error bars make it very hard to see the black and grey line trends. Consider altering the error bars in some way to allow the lines to become more visible. In the lower panel, why do the ELM simulation lines begin near 40 years instead of 0?*

We understand that Fig. 4 is difficult to interpret.
We will heed your suggestions when we remake Fig. 4.

- *Technical Corrections:*

  – *L10: remove 'when' from 'compared to when using...'*
  Done.

  – *L25: 'coupled' twice in this sentence, sounds awkward*
  We will rephrase this sentence.

  – *L34: need 'it' after 'implemented' here*
  We disagree, but reworded this sentence ("...and implemented..." → "..., implementing...") to possibly improve clarity.

  – *L225: the authors start what "with a baseline configuration. . ."?*
  We we mean something like "...start our presentation..." but this is an awkward way to begin a section.
  We will reword this sentence in our revised Section 4.

  – *L260: don't need comma after "ELM"*
  Removed.

  – *Figure 1: describe variable 'z' in the caption*
  We will rename this axis label "Depth".

  – *Figures 2 & 3: the bottom row of figures overlaps the y-axis units, space these out slightly so that each axis is legible.*
  We will space out the subplots accordingly.

  – *Figure 4: the top row of figures overlaps the y-axis units, space these out slightly so that the axis is legible.*
  We will space out the subplots accordingly.

  – *Figure 6: add description of density measurements are shown in blue, as well as that the dotted and dashed lines represent the characteristic density and pore close-off density, respectively, to the caption.*
  We will update the caption accordingly.

For a direct look at how we address these comments, please see the forthcoming revised manuscript.

Sincerely,

Adam M. Schneider et al.

---

## Author Comment (AC3) · 13 Nov 2020

13 Nov 2020

re: "Review of Schneider et al.", Anonymous Referee # 3, 16 Sep 2020

We appreciate you taking the time to express your concerns regarding our manuscript. We are taking them into great consideration as we revise. Following below is our response (in blue) to your enumerated concerns and specific comments, which are italicized here for reference:

1. *The first concern is about the statistical modelling approach explained in Sect 3.2. I didn't get very warm feelings about this. For instance, in the direct comparisons carried out in Sect. 4.1.1 and Figures 2-4, output from a coarse resolution ELM simulation with 6-hourly CRU-NCEP forcing is compared to steady-state profiles from the Herron & Langway model with idealized (synthetic) forcing. To me this feels like comparing apples to pears. Unnecessarily, it seems, since the ELM (like CLM) can also be forced with synthetic data in single-column mode, offering a more direct comparison. After all, to quote Arthern et al. (2010), "Changes in weather and climate can cause temperature, accumulation rate, and depositional density to vary. Consequently, and in violation of Sorge's Law, the density profile r(z, t) will fluctuate with time t." It is to be expected that this variation causes differences mainly in the upper ∼10 m of the firn pack, i.e. the active layer, which is unsurprisingly where the largest differences between the Herron&Langway model and the ELM simulations is found (Figure 2 &3). Arthern2010 further notes that "Alternative models, broadly based upon the Herron and Langway [1980] parameterization, have employed different formulations for the sensitivity to temperature [Li and Zwally, 2004; Helsen et al., 2008]". This quote is a hint that the temperature dependence of Herron & Langway is not to be taken as the truth, which is kind of what the authors seem to be doing in Section 4.1.2, but also in the next section (4.1.2) where they calibrate model coefficients using their synthetic HL density profiles. I feel the heavy reliance on the HL model and synthetic data isn't properly justified.*
   Regrettably, we failed to provide enough details regarding the statistical modeling. Also, it is apparent that our evaluation method, i.e., direct comparisons of ELM output to the empirical model of Herron and Langway (1980), does not provide a convincing assessment of improved model performance. Unfortunately, we are not able to force ELM in single column mode like in CLM.
   To remedy these shortcomings, first, we are expanding the description of our statistical methods (i.e., Section 3.2) used to calibrate model coefficients. Second we are subordinating our reliance on the direct comparisons to the model of Herron and Langway (1980) used to justify ELM improvements. Section 4 now includes a geographical analysis from which we calculate root mean squared errors (RMSE) with reference to the extensive SUMup (measurement) dataset described by Montgomery et al. (2018). Based on this analysis, we find that applying our (imperfect but useful) statistical calibration to the current CLM snowpack and firn densification model reduces RMSEs for the majority of model grid-cells covering the Greenland Ice Sheet (GrIS).

2. *My second concern is with the readability of the manuscript, and in particular the range of different model configurations that are presented, and the purpose for all of them. The*

*title of the paper is "Snowpack and firn densification in the E3SM", however at the end of the manuscript I'm lost to which results are now representative of the improved E3SM and which aren't. Line 184 seems to suggest that the coefficients in the new E3SM are optimised from one of the other models, however it isn't stated which one, and the final configuration in E3SM is not named. The wording in Line 320 ("might expect") and Line 376 ("a first step") is also contradictory to this, suggesting that the coefficient optimization isn't really applied at all in E3SM. Does that mean that the entire Section 4.1.2 is actually superfluous and could be removed? Are none of the models discussed in this paper actually adopted in E3SM? I encourage the authors to make this more explicit. Alternatively, the authors could choose to take the focus off E3SM and shape it into a more general firn-modelling paper, I'll leave that up to them.*

This concern highlights an issue with our presentation of experiments and the lack of clarity regarding the final E3SM implementation. As such, the reader becomes confused by the end and can be left with several important questions in addition to those posed above.

To elucidate the highly experimental nature of our study, we are revising Section 3 so that it will enumerate ELM cases that each have a specific firn model configuration. Section 4 then presents results of these cases in the same order that they were introduced in Section 3. We are also splitting Section 4 into separate "Results" and "Discussion" sections and are restricting our more speculative remarks regarding future model development.

3. *My third major concern is with the comparison to observed firn core data (Section 4.2). Here the authors aggregate ELM data from across the GrIS from a coarse resolution simulation, and compare this a single point measurement (or at least, an approximation to this). Again, this seems to me like comparing apples to pears and a pretty crude approach. For instance, there will most certainly be grid cells in the composite that experience melt, whereas the interest is on dry firn compaction. Can the ELM not be forced in single-column mode with high-resolution meteorological data from e.g. ERA5, or a high resolution run with E3SM, more approximate to the actual weather at the site?*

We originally relied on this observational comparison to correct obvious model biases because the measurements from Mosley-Thompson et al. (2001), which represent about 30 cores from various locations across the GrIS, exhibit much less variability than ELM density profiles throughout the same (approximate) domain of interest. Not only is this a crude approach, it also fails to provide a robust quantitative assessment of model realism that is missing in our justification of model improvement. Unfortunately, we are unable to force the ELM in stand-alone mode with higher resolution meteorological data. Given the current limitations, e.g., the low resolution ($\sim 2°$) forcing dataset and our one-dimensional snowpack model, however, our focus here is to reduce clear biases persisting globally in ELM when forced with plausible atmospheric reanalysis data. We believe that our experiments, while having limited capabilities, do show clear modeling errors that we attempt to correct with offline statistical modeling and validate with observations. In these comparisons, we exclude columns that undergo substantial melt by filtering out ELM grid-cells that have mean annual temperatures greater than -25 °C., as stated in the caption of Fig. 6.

Again, to improve our evaluation methodology, we are expanding the results (Section 4) to emphasize a new geographical analysis that compares hundreds of measurements from the SUMup dataset, described by Montgomery et al. (2018), to the nearest corresponding ELM

node in both space (via a location similarity matrix) and in time. This analysis stops short of a perfect "apples to apples" comparison, however, it does go further than our previous evaluation.

*All in all, I find this paper not convincing in its methodology, and I feel further justification or experiments are needed.*

We appreciate the constructive criticism and are working to address your valid concerns. We think you will find our expanded analysis more convincing and hope to provide more clarity in our forth-coming revision.

- *L61: pressure not pressures*
  We will correct.

- *L130: Actually, Muntjewerf et al. (2020) provides little detail on their model setup, and none on snow/firn modelling. Suggest to remove, and optionally replace with the following reference, which does actually provide more detail on snow modelling within CESM:*
  *van Kampenhout, L., Lenaerts, J. T. M., Lipscomb, W. H., Lhermitte, S., Noël, B., Vizcaíno, M., et al. (2020). Present-Day Greenland Ice Sheet Climate and Surface Mass Balance in CESM2. Journal of Geophysical Research: Earth Surface, 125(2), e2019JF005318.*
  *https://doi.org/10.1029/2019JF005318*
  We will update the reference accordingly.

- *L143: This title suggests that the new surface density scheme is only applied over ice sheets. However, this is not explained anywhere, so the title should be changed?*
  Perhaps you are correct. This sub-subsection title will be removed in our revision.

- *L180: Could you comment on what basis the values for T and A were selected? In Line 220 you define the "dry snow zone" as 0.5 m SWE / year, whereas here the value of 0.4 appears.*
  We selected the values based on Herron and Langway (1980). We reviewed their results to best determine where the model could be applied (in terms of mean annual temperature and accumulation rate) and focused our comparisons there.
  It is apparent that our basis needs to be better explained, probably in Section 3. In our revision, we are expanding the description of the statistical model, which relies heavily on the model of Herron and Langway (1980). Because we are subordinating the ELM experiment results to Herron and Langway (1980) comparison, we will add our basis for selecting temperature and accumulation domains in the expanded statistical modeling subsection. To add clarity, we will also comment on this domain selection in the discussion of the ELM experiment to Herron and Lanway (1980) comparisons.

- *L195: Could you comment on the quality of the meteorological data in CRU-NCEP over the regions of interest, i.e. ice sheets? Do you think the outcome of the simulations depends a great deal on the choice of meteorological forcing ?*
  The low resolution ($\sim 2°$) meteorological forcing provided by the CRUNCEP dataset is a primary limitation of our study. Because outcome of our ELM stand-alone simulations does, in fact, depend greatly on the surface boundary condition provided by the atmospheric forcing data, a mere geographical comparison of ELM to observations (which we are emphasizing in our revision) is not sufficient to disentangle density errors that are inherent from poor boundary conditions versus those resulting from bad firn densification parameterizations. This is why we originally fall back on a comparison to Herron and Langway (1980) for evaluation, i.e., because it controls for direct drivers of the fundamental processes (accumulation rate and temperature) that determine advective strain rates and resulting density profiles.

By emphasizing a geographical comparison to observations in the revised "Results" section (4) and by moving our comparison to Herron and Langway (1980) to a new "Discussion" section (5), our revision offers two perspectives. First, we provide a direct look at how ELM density profiles compare to the real world and evaluate ELM realism quantitatively by computing RMSEs with reference to the SUMup dataset (Koenig and Montgomery, 2019). Second, we also dig into how well the firn model configurations are processing whatever boundary conditions they inherent from the available atmospheric input, independent of how well such conditions are reproducing the local climate. The latter is particularly important to evaluate in preparation for future climate simulations during which a warming climate will result in new, drifting boundary conditions for a given ELM grid-cell. Such conditions at a fixed location are not represented in observations, large-scale reanalyses, or historical reconstructions.

- *L196: please specify what nominal resolution (in degrees or km) does the ne11 resolution correspond to?*
  The horizontal resolution of the ne11 grid is 2.8°.
  As requested, we will provide the nominal resolution when referring to ne11.

- *L215: Since this manuscript concerns a global model, and there are only 16 layers to begin with, two reference firn density profiles are probably justified. Just be aware that there are more firn cores out there, e.g. see Figure 1 in the recently published TC paper by Verjans et al. :*
  *Verjans, V., Leeson, A. A., Nemeth, C., Stevens, C. M., Kuipers Munneke, P., Noël, B., & van Wessem, J. M. (2020). Bayesian calibration of firn densification models. The Cryosphere, 14(9), 3017–3032. https://doi.org/10.5194/tc-14-3017-2020*
  Thank you for the positive remark and the following referral.
  We are actually expanding the comparison of ELM experiments to measurements (including performance metrics) using the SUMup dataset (Koenig and Montgomery, 2019). We elaborate on this new analysis in our responses above.

- *L260: remove comma after ELM ?*
  Removed.

- *L 373: near-surface firn densities that are too low, not large?*
  In A'76 (ELM v1), surface densities are too low. Just below the surface, densities are too high. This finding supports results from van Kampenhout et al. (2017) discovered in CLM version 4.
  We understand that the distinction here between "near-surface" and (e.g.) just "surface" is unclear. We will rephrase this and similar sentences throughout our revision to state more specifically where along the depth coordinate our findings apply.

- *L 376: E3SM Project: not clear, is this a reference?*
  Yes, this is a reference. We will change this to just "...within E3SM."

- *L 380: and AIS? Surface melt is believed (or known) to be important for the stability of ice*

*shelves.*
Ice shelves are not within the ELM domain and thus are not the focus of our present study.

- *Reference list: to avoid cluttering, I'd suggest to remove the URLs and replace with DOI where needed.*
Regarding the reference list, we will follow the guidelines and whatever recommendations given by the editorial support staff.

- *L 446 : fix title L 483 : fix title L 512 : fix title*
Thank you for bringing these fixes to our attention.
We will review (and fix) these titles in our revised manuscript.

- *Figure 4: The caption describes the meaning of the line graphs in the first row, but not the crosses used in the second row.*
We are revising Fig. 4.
We will also update the caption appropriately and add a description for the crosses.

- *Figure 5: The titles of the subfigures could be made more informative. Also, the colour bar could be made more restrictive, it appears densities $< 300 \ kg/m3$ do not occur at all.*
We will provide more informative titles and consider changing the color bar.

- *Figure 6: The legend appears not consistent with the previous figures, e.g. ELM-A'10 instead of just A'10.*
Our revised Sections 3 and 4 and our new "Discussion" Section (5) will be consistent in the labeling of ELM experiments.
We will also update all the figure legends accordingly.

For a direct look at how we address these concerns and comments, please see the forthcoming revised manuscript.

Sincerely,

Adam M. Schneider et al.